# Extrachromosomal DNA in the cancerous transformation of Barrett's oesophagus

Jens Luebeck[1,2], Alvin Wei Tian Ng[3,4], Patricia C. Galipeau[5], Xiaohong Li[6], Carissa A. Sanchez[5], Annalise C. Katz-Summercorn[3], Hoon Kim[7,8], Sriganesh Jammula[3], Yudou He[9,10,11], Scott M. Lippman[9], Roel G. W. Verhaak[12], Carlo C. Maley[13], Ludmil B. Alexandrov[9,10,11], Brian J. Reid[5,14,15], Rebecca C. Fitzgerald[3✉], Thomas G. Paulson[6✉], Howard Y. Chang[16,17✉], Sihan Wu[18✉], Vineet Bafna[1,19✉] & Paul S. Mischel[20,21✉]

Oncogene amplification on extrachromosomal DNA (ecDNA) drives the evolution of tumours and their resistance to treatment, and is associated with poor outcomes for patients with cancer[1–6]. At present, it is unclear whether ecDNA is a later manifestation of genomic instability, or whether it can be an early event in the transition from dysplasia to cancer. Here, to better understand the development of ecDNA, we analysed whole-genome sequencing (WGS) data from patients with oesophageal adenocarcinoma (EAC) or Barrett's oesophagus. These data included 206 biopsies in Barrett's oesophagus surveillance and EAC cohorts from Cambridge University. We also analysed WGS and histology data from biopsies that were collected across multiple regions at 2 time points from 80 patients in a case–control study at the Fred Hutchinson Cancer Center. In the Cambridge cohorts, the frequency of ecDNA increased between Barrett's-oesophagus-associated early-stage (24%) and late-stage (43%) EAC, suggesting that ecDNA is formed during cancer progression. In the cohort from the Fred Hutchinson Cancer Center, 33% of patients who developed EAC had at least one oesophageal biopsy with ecDNA before or at the diagnosis of EAC. In biopsies that were collected before cancer diagnosis, higher levels of ecDNA were present in samples from patients who later developed EAC than in samples from those who did not. We found that ecDNAs contained diverse collections of oncogenes and immunomodulatory genes. Furthermore, ecDNAs showed increases in copy number and structural complexity at more advanced stages of disease. Our findings show that ecDNA can develop early in the transition from high-grade dysplasia to cancer, and that ecDNAs progressively form and evolve under positive selection.

EAC is a highly lethal cancer that can arise from Barrett's oesophagus, a relatively common, pre-cancerous metaplastic condition that affects around 1.6% of the US population[7]. In addition to epidemiological and clinical features such as chronic gastro-oesophageal reflux disease, the age of the patient and the size of the Barrett's oesophagus lesion[8,9], changes in genomic copy number within Barrett's oesophagus lesions have been implicated in the development of EAC[7,10–15]. These changes include oncogene amplification, which frequently occurs on circular ecDNA particles (ecDNAs)[3]. ecDNAs are found in some of the most aggressive forms of cancer—including EAC, the highly accessible chromatin and altered *cis*- and *trans*-gene regulation of which enhance oncogenic transcriptional programs[16–20]. ecDNAs lack centromeres, and are consequently subject to random inheritance during cell division, driving intratumoral genetic heterogeneity[6]. These unique features of ecDNAs contribute to aggressive tumour growth, accelerated evolution and resistance to treatment. Patients with ecDNA-containing tumours have significantly shorter survival, even compared to other forms of genomic focal amplification[3]. Computational tools can detect ecDNA in WGS data from biopsies[21–23]. However, the relative paucity of pre-cancer-to-cancer longitudinal studies, together with the challenges

[1]Department of Computer Science and Engineering, University of California at San Diego, La Jolla, CA, USA. [2]Bioinformatics and Systems Biology Graduate Program, University of California at San Diego, La Jolla, CA, USA. [3]Early Cancer Institute, Hutchison Research Centre, University of Cambridge, Cambridge, UK. [4]Cancer Research UK Cambridge Institute, University of Cambridge, Cambridge, UK. [5]Divisions of Human Biology and Public Health Sciences, Fred Hutchinson Cancer Center, Seattle, WA, USA. [6]Clinical Research Division, Fred Hutchinson Cancer Center, Seattle, WA, USA. [7]Department of Biopharmaceutical Convergence, Sungkyunkwan University, Suwon, Republic of Korea. [8]Department of Biohealth Regulatory Science, Sungkyunkwan University, Suwon, Republic of Korea. [9]Moores Cancer Center, UC San Diego Health, La Jolla, CA, USA. [10]Department of Cellular and Molecular Medicine, University of California at San Diego, La Jolla, CA, USA. [11]Department of Bioengineering, University of California at San Diego, La Jolla, CA, USA. [12]The Jackson Laboratory for Genomic Medicine, Farmington, CT, USA. [13]Biodesign Institute, Arizona State University, Tempe, AZ, USA. [14]Department of Genome Sciences, University of Washington, Seattle, WA, USA. [15]Department of Medicine, University of Washington, Seattle, WA, USA. [16]Center for Personal Dynamic Regulomes, Stanford University, Stanford, CA, USA. [17]Howard Hughes Medical Institute, Stanford University, Stanford, CA, USA. [18]Children's Medical Center Research Institute, University of Texas Southwestern Medical Center, Dallas, TX, USA. [19]Halıcıoğlu Data Science Institute, University of California at San Diego, La Jolla, CA, USA. [20]Department of Pathology, Stanford University School of Medicine, Stanford, CA, USA. [21]Sarafan Chemistry, Engineering, and Medicine for Human Health (Sarafan ChEM-H), Stanford University, Stanford, CA, USA. ✉e-mail: rcf29@mrc-cu.cam.ac.uk; tpaulson@fredhutch.org; howchang@stanford.edu; Sihan.Wu@UTSouthwestern.edu; vbafna@ucsd.edu; pmischel@stanford.edu

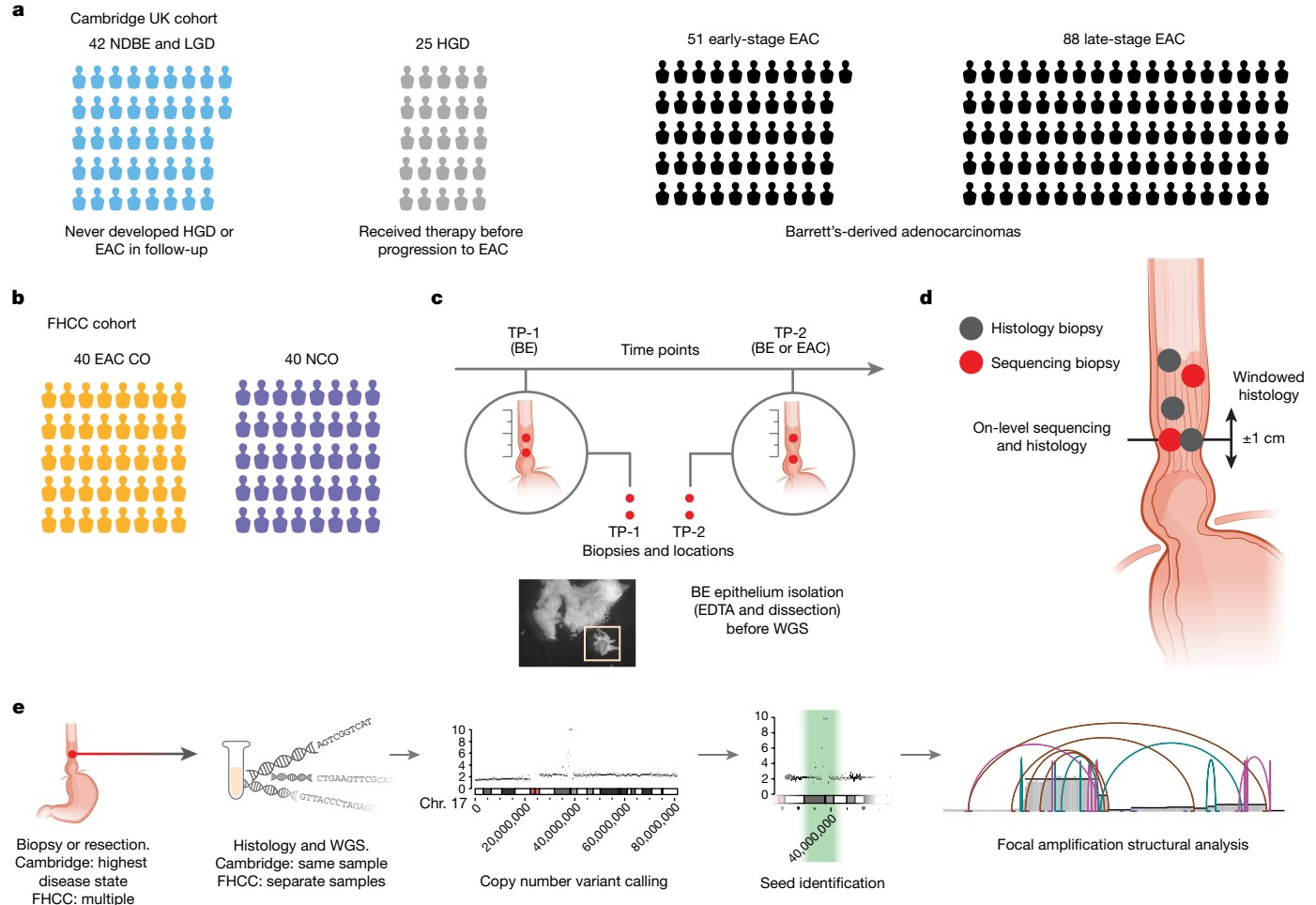

**Fig. 1 | Study and analysis designs. a**, Breakdown of the histological disease states among patients with Barrett's oesophagus in the Cambridge selected cross-sectional study, representing the highest disease state for that patient. NDBE, non-dysplastic Barrett's oesophagus. **b**, The FHCC cohort consisted of 80 patients for whom biopsies were collected prospectively. The cohort was separated later into two groups of 40 patients who had cancer outcomes (CO) and non-cancer outcomes (NCO). **c**, Sample collection at time points TP-1 and TP-2 for sequencing biopsies and histology biopsies. Two sequencing biopsies were collected at each time point. Before sequencing, ethylenediaminetetraacetic acid (EDTA) application and microdissection were performed to isolate Barrett's oesophagus (BE) tissue and improve purity for sequencing. Highlighted box indicates isolated Barrett's oesophagus tissue (box width indicates approximately 50 μm). **d**, WGS biopsies and histology biopsies were collected independently. Some histology and sequencing biopsies were taken at the same level of the oesophagus (on-level), and some histology biopsies fell within a ±1-cm window of the measured height of the sequencing biopsy (windowed histology). **e**, Experimental workflow for analysing the WGS samples. A brief overview of the process by which biopsies were selected, sequenced and characterized by AmpliconArchitect.

of interpreting clonality in the face of non-Mendelian genetics, have made it difficult to determine whether ecDNAs arise early in tumorigenesis and contribute to the transformation of dysplasia into cancer. Previous reports have hypothesized or reported the existence of ecDNA in pre-cancer samples derived from individuals with Barrett's oesophagus[11,14,24,25] suggesting that ecDNA has a role in the malignant transformation to EAC[11]. Two surveillance studies of patients with Barrett's oesophagus, including a longitudinal case–control study with multi-regional WGS sampling, and a completely independent, cross-sectional surveillance cohort, with full histological correlatives, provided us with an opportunity to study the role of ecDNA in the transition from Barrett's oesophagus to EAC.

## Study samples

We analysed WGS data from a Cambridge University cross-sectional surveillance cohort of 206 patients with biopsy-validated Barrett's oesophagus (Supplementary Table 1). This Cambridge cohort included 42 patients with non-dysplastic Barrett's oesophagus or low-grade dysplasia (LGD) who never developed high-grade dysplasia (HGD) or EAC during follow-up; 25 patients with HGD; 51 patients with early-stage (stage I) EAC; and 88 patients with late-stage (stage II–IV) EAC (Fig. 1a) using the American Joint Committee on Cancer (AJCC) staging system[26]. Histology and WGS sequencing were performed on the same biopsies (Methods, 'Cambridge sample selection'). We also analysed 20 EAC tumours from The Cancer Genome Atlas (TCGA) oesophageal carcinoma study[27] (ESCA), composed of 6 early-stage and 14 late-stage tumours.

We analysed WGS data from oesophageal biopsies collected in an independent, prospectively collected case–control study conducted at the Fred Hutchinson Cancer Center (FHCC) of patients with Barrett's oesophagus[14] (Fig. 1b), including 40 patients with a cancer outcome and 40 patients who did not develop cancer (non-cancer outcome) during the study period or during follow-up (mean: 10.5 years; Supplementary Table 2). At least two biopsies for WGS were obtained by isolating epithelial tissue (Fig. 1c) from the Barrett's oesophagus at each of the two primary study time points–time point 1 (TP-1) and time

point 2 (TP-2)—which were, on average, 2.9 years and 3.4 years apart for patients with a cancer outcome and those with a non-cancer outcome, respectively (Supplementary Table 1). Histology samples were also collected independently of the sequencing biopsies, including from the same, or close to the same, level in the oesophagus (Fig. 1d). At TP-2, biopsies from the same level of the oesophagus (or as close as possible) as the EAC were used for sequencing (Methods, 'FHCC cohort histology'). For the resected tumour from patient 391, sequencing was also available. We applied the AmpliconArchitect method for ecDNA detection (Fig. 1e and Methods, 'ecDNA detection and characterization'), after identifying seed regions of possible focal amplifications in copy number calls. The resulting genome graphs described the fine structure of the amplicon, and explorations of those graphs were subsequently classified by AmpliconClassifier as specific amplicon types (Supplementary Table 3 and Supplementary Information).

## ecDNA and Barrett's oesophagus

ecDNA was not detected in any of the non-dysplastic Barrett's oesophagus samples or any of the LGD samples in the cross-sectional Barrett's oesophagus surveillance Cambridge cohort (Extended Data Fig. 1a). By contrast, ecDNA was found in tumours from 13 out of 51 patients (25%) with early-stage (stage I) EAC and in tumours from 38 out of 88 patients (43%) with late-stage tumours (stage II–IV) (Fig. 2a). The occurrence of ecDNA was significantly enriched in early-stage EAC versus non-dysplastic Barrett's oesophagus or LGD (Fig. 2b; Fisher's exact test, $P = 1.8 \times 10^{-4}$, one-sided) with an increased frequency of ecDNA in late-stage compared to early-stage tumours (Extended Data Fig. 1b; odds ratio = 2.2, confidence interval = 1.0–4.7, Fisher's exact test, $P = 0.027$, one-sided). ecDNA was detected in a nearly identical fraction of an independent cohort of late-stage EAC tumours from TCGA (6 out of 14 tumours; 43%).

The FHCC study incorporated multi-regional, longitudinal sampling from before and at cancer diagnosis. We examined the development of ecDNA over time in biopsies from patients with Barrett's oesophagus tissue that progressed to an EAC end-point versus those patients with Barrett's oesophagus tissue that remained benign at the highest detectable disease state. At cancer diagnosis (TP-2), ecDNA was detected in samples from 11 out of 40 patients with a cancer outcome who developed EAC (28%) (Fig. 2c and Extended Data Fig. 2), consistent with the 25% frequency of ecDNA that was found in the Cambridge cohort of patients with early-stage cancer. ecDNA was detected in biopsies from only one out of 40 patients with a non-cancer outcome (Extended Data Fig. 3). Notably for the non-cancer-outcome ecDNA biopsies, *KRAS* was amplified (Extended Data Fig. 4); however, the patient died of causes unrelated to Barrett's oesophagus 2.84 years after TP-2.

We also analysed 20 long-term follow-up samples collected from 10 patients with a non-cancer outcome (median 9.6 years after TP-2) with Barrett's oesophagus tissue that maintained non-dysplastic Barrett's oesophagus or LGD status and remained ecDNA-negative (Extended Data Fig. 5a). The median duration of follow-up was 10.5 years for the FHCC patients with a non-cancer outcome, with 85% being followed for more than 5 years (Supplementary Table 2 and Extended Data Fig. 5b). Furthermore, in both time points together, ecDNA was found in samples from one out of 40 patients with a non-cancer outcome and in samples from 13 out of 40 patients with a cancer outcome (Fig. 2d), showing a highly significant association between ecDNA in Barrett's oesophagus biopsies and progression to EAC (Fisher's exact test, $P = 3.3 \times 10^{-4}$, one-sided).

## ecDNA can arise in high-grade dysplasia

The design of the longitudinal case–control FHCC study enabled the timing of ecDNA development in Barrett's oesophagus segments to be determined in patients with a cancer outcome. Notably, ecDNA was found at TP-1, before the development of cancer, in biopsy tissues from

7 out of 40 patients (18%), and all 7 of these individuals subsequently developed EAC (Fig. 2d). In addition, at TP-1, HGD was detected in at least one histology biopsy for 27 out of 40 patients (67.5%). Six TP-1 samples that contained ecDNA could be matched to an on-level histology biopsy, all showing HGD (Fig. 2e). By contrast, 46% (21 out of 46) of the ecDNA-negative TP-1 sequencing biopsies could be matched to on-level HGD, indicating a significant association of ecDNA and HGD in the pre-cancer samples (Fig. 2e; Fisher's exact test, $P = 0.015$, one-sided).

In samples from patients with a cancer outcome that were collected at TP-2, when cancer was first diagnosed, we associated 54 sequencing biopsies to on-level histology. ecDNAs were identified in 11 of these sequencing biopsies, 9 of which (82%) were associated with on-level EAC, with the remaining 2 being associated with on-level HGD (Fig. 2f). By contrast, among the remaining 43 ecDNA-negative biopsies, only 20 out of 43 (47%) were associated with on-level EAC, with the remaining 23 (53%) being associated with on-level Barrett's oesophagus or HGD (Fig. 2f; Fisher's exact test, $P = 0.037$, one-sided). The specificity of the association of ecDNA with a worsened pathological status at both time points suggests that ecDNAs are enriched in Barrett's oesophagus clones that become cancer. In the Cambridge cohort, ecDNA was detected in only one of the 25 patients with HGD (Extended Data Fig. 1a). However, in that cohort, HGD was treated immediately after detection, so it was not possible to determine whether the HGD samples would subsequently have progressed to cancer.

## *TP53* alterations and ecDNA formation

We analysed a number of properties related to the samples in the context of ecDNA, ranging from purity and ploidy (Supplementary Table 4 and Supplementary Fig. 4) to other genomic features, such as *TP53*. Disruption of *TP53* enables genomic instability[13,14,28,29], and we found a strong association in both FHCC and Cambridge samples between the *TP53* alteration status (Methods, '*TP53* alteration analysis') and the presence of ecDNA (Extended Data Fig. 6a,b). All eight FHCC samples in which ecDNA was found before cancer diagnosis (TP-1) showed biallelic disruption of *TP53*. The appearance of ecDNA as a subset of *TP53*-altered cases suggests that the prior loss of *TP53* enables ecDNA formation.

Whole-genome duplication (WGD) and chromothripsis are tied closely to genome instability[11,30–33], and those mechanisms might contribute to ecDNA formation[11,15,30–32,34,35]. In the FHCC samples, we found that WGD and chromothripsis were significantly associated with *TP53* alteration (Extended Data Fig. 6c,d), indicative of its role in mediating genomic stability. However, many of the samples with ecDNA did not show evidence of chromothripsis or WGD (Extended Data Fig. 6e,f), indicating that there are other mechanisms of ecDNA formation after *TP53* alteration.

## ecDNA and malignant transformation

To better understand the potential relationship between ecDNA and the transition from HGD to EAC, we studied an individual patient in the FHCC cancer-outcome cohort (patient 391), whose WGS data were collected at four endoscopies over a seven-year period (Fig. 3a). At first, HGD was detected at two different locations within the Barrett's oesophagus segment. Chromosomal *ERBB2* amplification through breakage–fusion–bridge (BFB) cycles (Extended Data Fig. 7a) and *TP53* alterations were present in these biopsies (Fig. 3b). An ecDNA (ecDNA-1), containing *AP2B1*, *GAS2L2* and *RASL10B*, was only detected (Fig. 3b,c and Extended Data Fig. 7b) after 5.6 years. The lesions did not progress to EAC for another 6.5 months, at which point a second ecDNA (ecDNA-2) containing *SOCS1*, *CIITA* and *RMI2* was detected (Fig. 3b,c and Extended Data Fig. 7c). *SOCS1* is a suppressor of cytokine signalling, including interferon-γ[36], which may foster escape from cytotoxic T cells[37]. *CIITA* is an immunomodulatory master transcription factor for

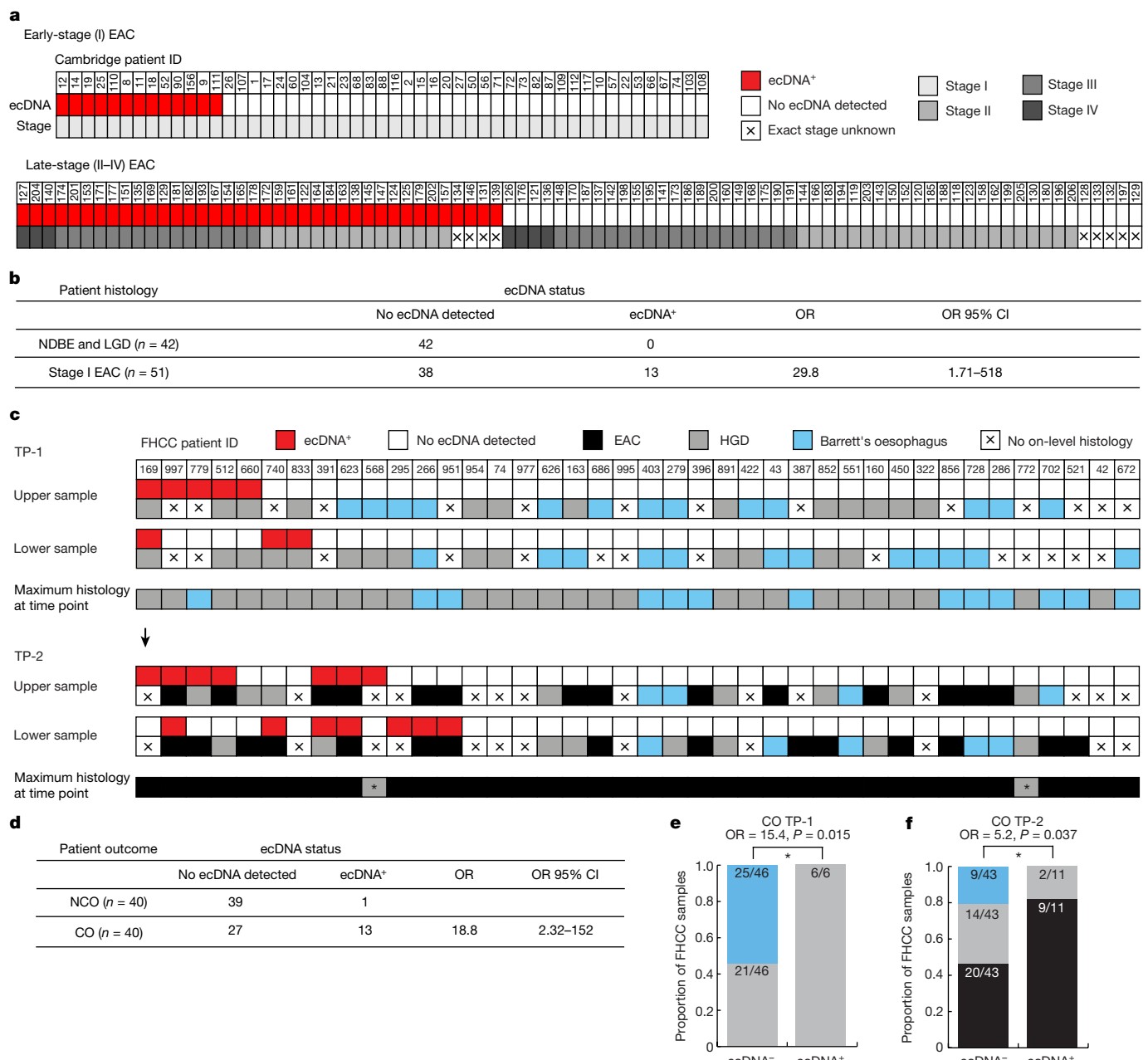

**Fig. 2 | Association of ecDNA with histology. a**, Characterization of the ecDNA status and cancer stage of patient samples from the Cambridge cohorts of patients with early- and late-stage EAC. **b**, Comparison of the ecDNA status and histological group of samples reveals an association between ecDNA and early-stage EAC. The odds ratio (OR) and the confidence interval (CI) of the OR are shown. **c**, Characterization of the ecDNA status and on-level histology of samples collected for FHCC CO patients across time points TP-1 and TP-2 for the two oesophageal sequencing samples ('upper' and 'lower'). The maximum histology of any biopsy from that time point is also shown. Asterisk indicates cancer diagnosis made at next endoscopy (1.44 and 8.16 months after TP-2 for patients 568 and 772, respectively). **d**, Comparison of ecDNA status in any

FHCC patient sample and cancer-outcome status among patients reveals an association between ecDNA and cancer outcome. **e**, Among FHCC CO patients, the proportion of TP-1 samples without HGD or EAC in on-level histology (having Barrett's oesophagus or LGD) versus with HGD in the on-level histology, separated by ecDNA status, shows an enrichment for ecDNA with advanced disease status (Fisher's exact test, one-sided). **f**, Among FHCC CO patients, the proportion of TP-2 samples without EAC in on-level histology (having HGD or Barrett's oesophagus) versus with EAC in on-level histology, separated by ecDNA status, shows an association between ecDNA and the development of EAC (Fisher's exact test, one-sided).

antigen presentation[38], and its translocation is immunosuppressive[39]. *RMI2* is a component of the Bloom helicase complex, which is involved in homologous recombination and might have a role in lung cancer metastases[40]. A subsequent surgical resection of the tumour confirmed both ecDNA-1 and ecDNA-2, whereas the tissue that contained only ecDNA-1, *TP53* alteration and chromosomal *ERBB2* amplification remained HGD. These results suggest that multiple and ongoing focal

amplification events occur in dysplastic tissues[41,42], enhancing the fitness of a clone during malignant transformation.

## ecDNAs with common origins

To compare the fine structures of ecDNAs across multiple time points and biopsies from the same individual, we developed an amplicon

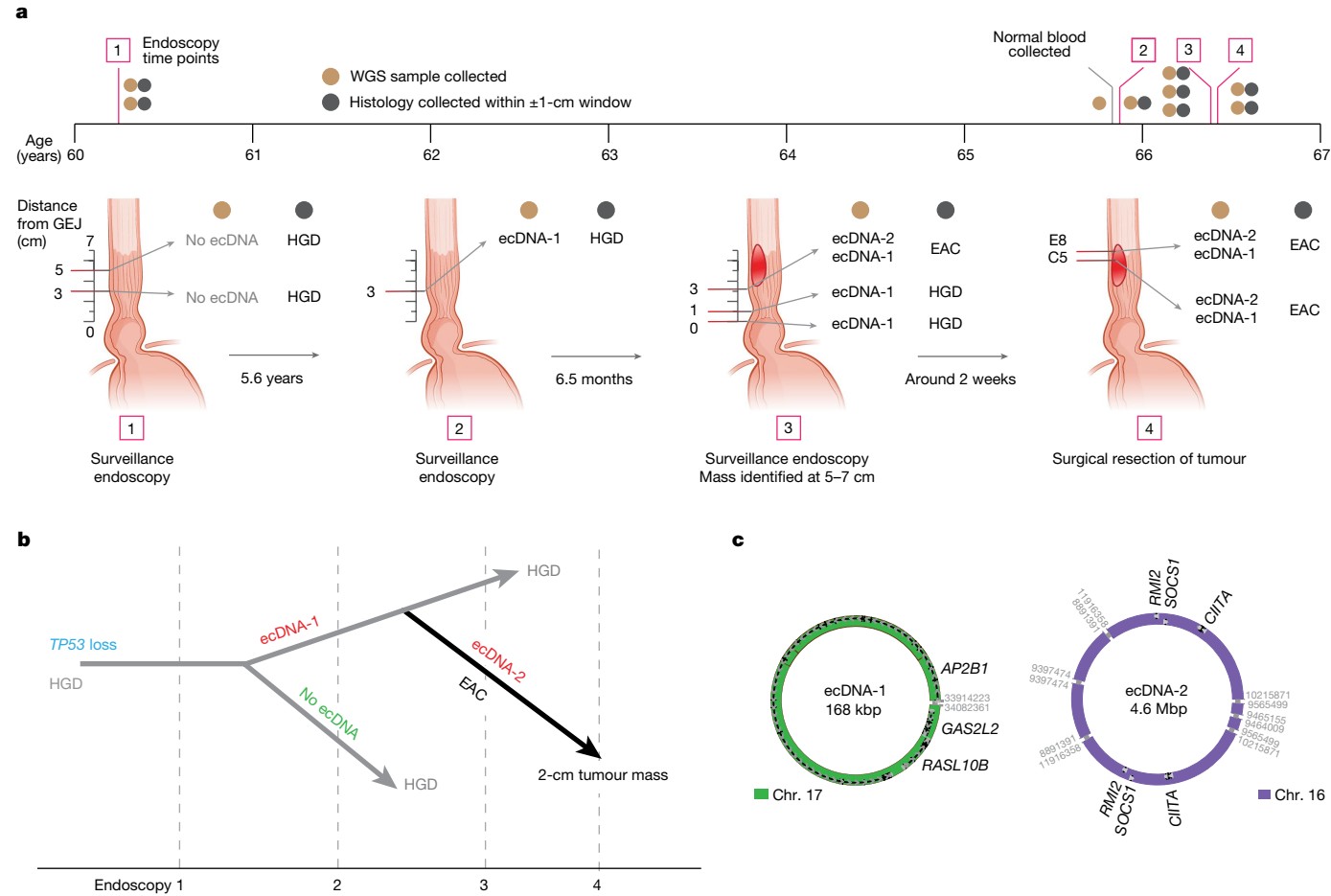

**Fig. 3 | ecDNA and the malignant transformation. a**, Timeline of sample collection for FHCC CO patient 391 relative to patient age. Summary of the ecDNA status and windowed-histology status for four endoscopies, with the time interval between each indicated. The distance of the biopsy from the gastro-oesophageal junction (GEJ) is also shown. The two resection samples are labelled as E8 and C5. Two distinct species of ecDNA are labelled as ecDNA-1 and ecDNA-2. **b**, Inferred phylogeny of Barrett's oesophagus samples from patient 391 across the four endoscopies, starting from *TP53* alteration, with branching reporting the ecDNA formation events, annotated by the histological status of the sample (windowed). **c**, Left, structure of ecDNA-1, first detected in endoscopy 2, in which HGD was detected within ±1 cm. Right, structure of ecDNA-2, first detected in endoscopy 3, in which EAC was diagnosed and present within ±1 cm.

similarity score ranging from 0 to 1 (Supplementary Fig. 1a–e and Supplementary Information). Genomically overlapping ecDNAs in different samples from the same patient showed a high similarity, consistent with a common origin (Supplementary Fig. 1e). All ten genomically overlapping ecDNA pairs from within the same FHCC individuals, reidentified between different biopsies, showed a significant similarity ($P < 0.05$) (Supplementary Table 5). Thus, ecDNAs detected in pre-cancer are frequently maintained through the transition to cancer, and genomically overlapping ecDNAs identified from multi-region sampling are likely to have a common origin. Together, these data suggest that ecDNA can be a truncal event in the formation and evolution of EAC.

## Selection and evolution of ecDNAs

We detected a marked increase in the frequency of ecDNA in biopsies from patients with a cancer outcome before clinical detection of cancer, and even higher levels of ecDNA in biopsies or resections from later-stage cancers (Fig. 4a). To better understand these observations, we characterized 137 ecDNAs across all samples from 75 patients with ecDNA-positive Barrett's oesophagus, Barrett's-oesophagus-derived HGD or Barrett's-oesophagus-adjacent EAC. ecDNA copy number was significantly higher in EAC samples than in pre-cancer samples (Fig. 4b; Mann–Whitney $U$ test, $P = 0.033$, one-sided). Although the lengths of genomic regions captured on ecDNA were not significantly different between pre-cancer and EAC states (Extended Data Fig. 8; Mann–Whitney $U$ test, $P = 0.44$, two-sided), the complexity of structural rearrangements in ecDNA-derived regions increased between pre-cancer and EAC (Fig. 4c; Mann–Whitney $U$ test, $P = 0.043$, one-sided), suggesting a significant increase in the heterogeneity of ecDNA structures with the evolution of tumours. We next investigated copy number changes in eight pairs of clonal ecDNA in which the same ecDNAs (on the basis of amplicon similarity score) appeared in different sequencing biopsies from the same patient, and for which the biopsies also had windowed-histology data available (Supplementary Table 1). When both ecDNA occurrences were associated with the same histology, the ecDNA copy numbers were highly similar. However, if one sample was associated with a more severe histological status than the other, the ecDNA copy number was significantly higher in that sample (Fig. 4d; Mann–Whitney $U$ test, $P = 0.029$, one-sided). These data suggest that ecDNA confers a strong selective advantage to the Barrett's oesophagus clones that eventually progress to EAC, and that pre-cancer ecDNAs are subject to continued evolution during malignant transformation and progression, leading to increased heterogeneity and copy number.

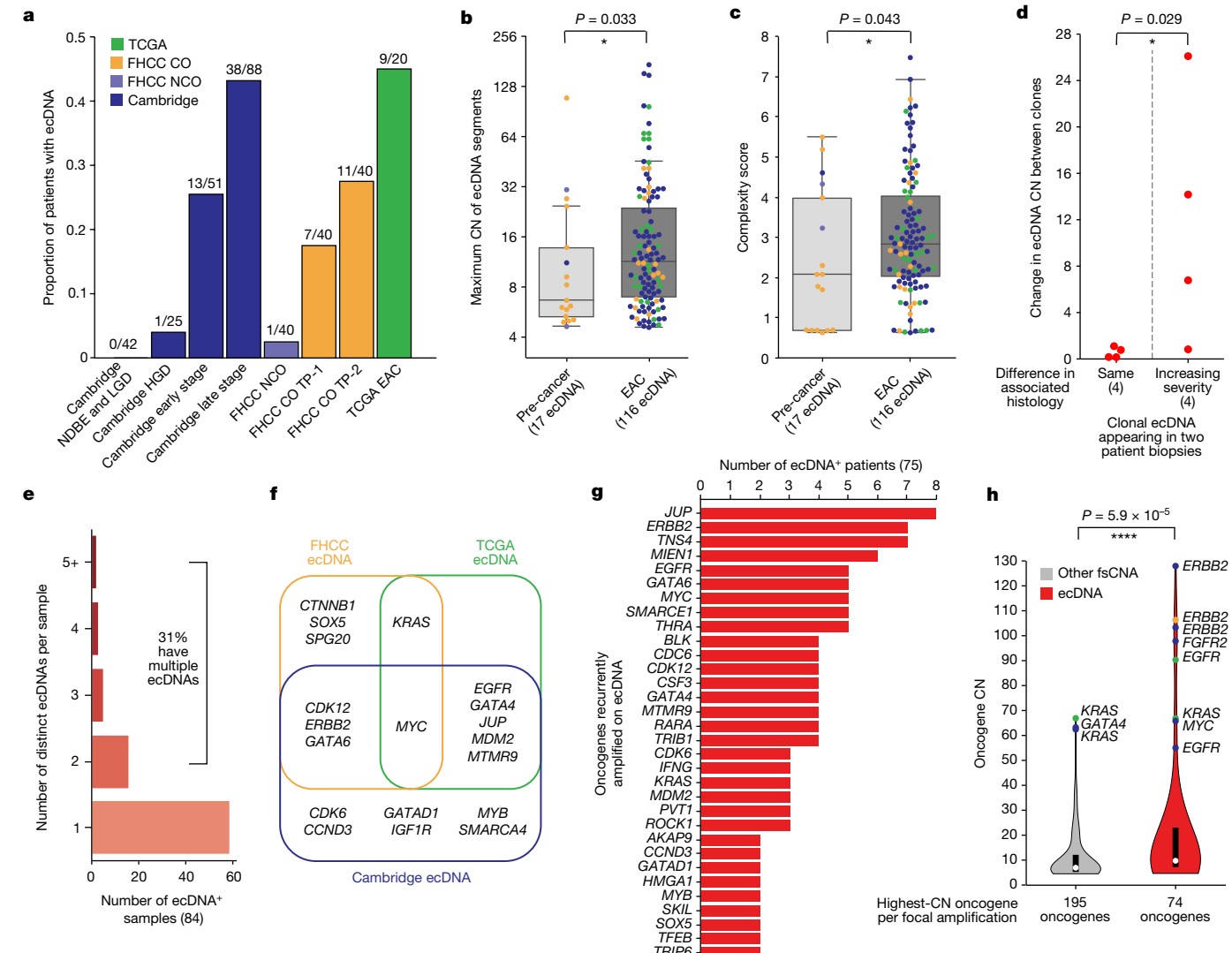

**Fig. 4 | ecDNA properties. a**, Proportion of patients with ecDNA detected in any sample across all study cohorts. **b**, Maximum genomic copy number (CN) of ecDNA segments in pre-cancer samples and EAC (or EAC-linked for FHCC) samples, coloured by sample study source. **c**, Complexity score of focally amplified ecDNA-positive genomic regions for pre-cancer and EAC samples. **d**, For ecDNAs identified across multiple FHCC samples that were determined to be clonal on the basis of amplicon similarity, the increase in ecDNA copy number for each pair of clonal ecDNAs, separated by the difference in associated histology of the two samples, shows an association between increasing copy number and increasing histological severity. **e**, Number of distinct ecDNAs per sample, among the ecDNA-positive samples, from all combined sources of data. **f**, Comparative overlap of Barrett's-oesophagus-associated oncogenes found on ecDNA in the four cohorts. **g**, For oncogenes recurrently detected on ecDNA in samples from different patients, the number of patients with a sample that has the listed oncogene included on ecDNA. **h**, Oncogene copy number for the focally amplified oncogene with the highest copy number on each unique focal amplification (ecDNA or non-ecDNA fsCNA) is significantly higher on ecDNA versus non-ecDNA fsCNA.

Twenty-six out of 83 (31%) ecDNA-positive samples from the combined study sources contained more than one species of ecDNA (Fig. 4e), enabling multiple oncogene amplifications. Multiple species of ecDNA could also be detected in Barrett's oesophagus HGD samples from patients who progressed to EAC (Supplementary Table 3), raising the possibility that tumours might achieve subclonal ecDNA heterogeneity early on, and that competition between multiple distinct ecDNAs could have a role in the evolution of EAC.

## Diversity of ecDNA-borne genes

We identified a large diversity of oncogenes on the ecDNAs, many of which were not detected on non-ecDNA focal amplifications (Extended Data Fig. 9a). ecDNAs were significantly enriched for oncogenes as compared to non-oncogenes (Extended Data Fig. 9b; Fisher's exact test,

$P = 8.9 \times 10^{-4}$, one-sided)—including oncogenes that are known to drive EAC, such as *ERBB2*, *KRAS* and *MYC*, which were recurrently detected on ecDNAs found in Barrett's oesophagus and EAC across multiple cohorts (Fig. 4f,g and Supplementary Table 6). Furthermore, 33.1% of the ecDNAs contained multiple oncogenes on the same molecule (Extended Data Fig. 9c). ecDNAs contained 0.76 unique oncogenes per amplicon (97 oncogenes in 127 ecDNAs), compared to 0.52 (192/373) unique oncogenes per amplicon in non-extrachromosomal focal somatic copy number amplifications (fsCNAs), suggesting that ecDNA may allow a wider variety of oncogene amplifications.

ecDNA amplification was associated with a greater maximum oncogene copy number than other fsCNAs (Fig. 4h; distribution mean copy number = 11.6 and 21.3 for non-ecDNA and ecDNA, respectively, Mann–Whitney $U$ test, $P = 5.9 \times 10^{-5}$, one-sided), with some ecDNA genes surpassing 100 copies. ecDNA also permitted greater diversity in

maximum copy number than did non-ecDNA fsCNA (copy number variance = 687.9 versus 122.2 in non-ecDNA fsCNA, Levene's test, $P = 1.5 \times 10^{-4}$). Notably, many ecDNA genes (79 in total) were associated with immunomodulation (Supplementary Table 7 and Extended Data Fig. 10a), and only 25 of the 79 were already present in the set of canonical oncogenes. The ecDNA amplified immunomodulatory genes had a significantly higher copy number compared to those on other fsCNAs (Extended Data Fig. 10b; Mann–Whitney $U$ test, $P = 4.1 \times 10^{-3}$, one-sided).

A comparison of genomic regions that are predicted to be on ecDNA and oncogene intervals that are known to associate specifically with Barrett's oesophagus and EAC (Supplementary Table 8 and Supplementary Fig. 2) showed a statistically significant overlap (ISTAT test[43], $P = 3.1 \times 10^{-5}$; Supplementary Information), suggesting that—despite the high diversity of ecDNA-borne oncogenes—ecDNAs are positively selected in a manner that is specific to cancer type.

## Discussion

Oncogene amplification on ecDNA enables tumours to evolve at an accelerated rate, which drives rapid resistance to therapy and contributes to shorter survival for patients[2,3,5]. It has been unclear whether ecDNA can contribute to the transformation of pre-cancer to cancer, or whether it is a later manifestation of tumour genomic instability. Here, in multiple cohorts of patients with Barrett's oesophagus, we show that ecDNA appears in HGD, and that its presence is strongly associated with EAC progression.

Typical phylogenetic approaches to track cancer clonality assume chromosomal inheritance. In consequence, it has been challenging to infer the clonality and evolution of ecDNA-driven cancers. Our results show that in the evolution of tumours from pre-cancer to cancer, ecDNA confers a strong selective advantage to the Barrett's oesophagus clones that eventually progress to EAC. The substantial heterogeneity in ecDNA-containing cancers might promote rapid and frequent branching of the phylogenetic tree, fostered by the non-chromosomal inheritance of ecDNA during cell division. Moreover, the increased prevalence and complexity of ecDNA structures in oesophageal cancer samples suggests ongoing selection and evolution during the formation and progression of tumours[44].

Our results strongly suggest that ecDNAs usually arise in regions of HGD in patients with Barrett's oesophagus, and nearly always in the context of TP53 alteration. These results complement previous reports that found that TP53 alteration and altered copy number might drive the transition from metaplasia to dysplasia[10,13,14,45,46], showing the cooperative nature of various genetic and epigenetic alterations, and suggesting that ecDNA formation represents a particularly potent driver of transformation and could be an opportunity for specific therapeutic intervention. Moreover, our analysis of a single patient across time supports and extends a report[46] suggesting that TP53 alteration results in polyploid tumours with multiple amplified or gained regions providing a reservoir for amplifying oncogenes.

Freed from Mendelian constraints, ecDNA amplifies a broader range of oncogenes, and their copy numbers increase rapidly and markedly in EAC, consistent with strong positive selection. Increased ecDNA heterogeneity may also enhance adaptation to changing conditions. Notably, the clonal selection and maintenance of immunomodulatory genes on ecDNA before cancer development could aid immune evasion. Together, these results indicate that ecDNA contributes to the development of cancer through several mechanisms. These findings shed light on how ecDNA can arise before the development of full-blown cancer, indicating that it is not simply a late manifestation of genome instability, and raise the possibility of earlier intervention or prevention for patients with ecDNA-containing tumours.

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

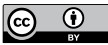

## Methods

### ecDNA detection and characterization

DNA copy number alterations were detected using CNVKit (ref. 47; FHCC and TCGA samples) and ASCAT (ref. 48; Cambridge samples). AmpliconSuite-pipeline (v.0.1203.12) was used to identify candidate seed regions for the detection and characterization of ecDNA using AmpliconArchitect v.1.2 (ref. 21) and AmpliconClassifier (v.0.4.13) (Supplementary Information and Fig. 1e), and circular visualizations of candidate ecDNA structures were generated using CycleViz (v.0.1.1). Amplicon complexity score was computed on the basis of the diversity of the amplicon structure decompositions output by AmpliconArchitect (Supplementary Information and Supplementary Fig. 3).

### Statistical analysis

We used SciPy v.1.9.1 (ref. 49) to perform all statistical tests in the study, with the exception of the ecDNA region–oncogene overlap significance test, for which we used ISTAT v.1.0.0 (ref. 43 and Supplementary Information). When computing odds ratios, if any cell in the two-by-two table was zero, the Haldane correction[50] was applied to every cell in the table. The significance of odds ratios and differences in event frequencies between groups were assessed by Fisher's exact test. The default test type was two-sided in statistical tests, unless otherwise specified. For data represented in box plots, the centre line is the median, the box limits are the upper and lower quartiles and the whiskers are 1.5 times the interquartile range, or represent minimum or maximum values if there are no outliers.

### Cambridge sample selection

We identified suitable patients from a prospective surveillance study of more than 3,000 patients with pre-cancerous lesions in a previous study[13], in which the follow-up and methods for pathology reviewed were described. There was a minimum follow-up of 44 months (median 139, maximum 258) for the cohorts that did not progress to HGD or EAC, and a median follow-up of 57 months (range 0–249) for the dysplastic cohort. We analysed WGS data from 206 patients in cross-sectional Barrett's oesophagus surveillance Cambridge cohorts with biopsy-validated Barrett's oesophagus, including 27 patients with non-dysplastic Barrett's oesophagus, 15 patients with LGD who never developed HGD or EAC during follow-up, 25 patients with HGD, 51 patients with early-stage EAC (AJCC stage I) and 88 patients with late-stage EAC (AJCC stage II–IV (Fig. 1a). Patients with low-grade Barrett's oesophagus and high-grade Barrett's oesophagus underwent surveillance at Cambridge University Hospitals NHS Trust and consented prospectively to a biomarker and genomic characterization study (Cell Determinants Biomarker, REC no. 01/149, BEST2 REC no. 10/H0308/71).

Patients included in the Cambridge surveillance cohorts were all treatment-naive—that is, the patients had received no neoadjuvant chemotherapy, radiation or ablation therapies—except for two of the patients with late-stage EAC (patients 167 and 155), who had received previous therapy, as detailed in Supplementary Table 1 and Extended Data Fig. 1a.

Strict selection criteria were implemented to ensure that only the highest-cellularity biopsies, with agreement of histological grade, were sequenced. Potential biopsies were placed into optimal cutting temperature compound and a single section was cut and stained with haematoxylin and eosin (H&E). These were reviewed by at least two consultant pathologists to assess the composition of the biopsy. All pathologists were blinded to the grade of the patient. Samples with no agreement were reviewed by a third pathologist to reach a consensus. Dysplastic samples for sequencing had to have a pathological cellularity for dysplasia of at least 30% and were labelled to be consistent with the highest pathology grade reported within the biopsy (tumour cellularity of 70% or higher for early-stage cancers). Non-dysplastic Barrett's oesophagus biopsies had to contain intestinal metaplasia.

Patients with early- and late-stage EAC in the Cambridge cohorts were recruited for the EAC International Cancer Genome Consortium (ICGC) study, for which samples were collected through the UK-wide Oesophageal Cancer Classification and Molecular Stratification (OCCAMS, REC no. 10-H0305-1) consortium. For early-stage cancer samples, samples with a cellularity of 70% or higher were included, consistent with ICGC guidelines. Ethical approvals for these trials were from the East of England–Cambridge Central Research Ethics Committee. EAC samples were prospectively collected as endoscopic biopsies or resection specimens. All tissue samples were snap-frozen and blood or normal squamous epithelium (at least 5 cm from the tumour) was used as a germline reference as previously described[13].

Barrett's oesophagus research samples were collected at every 2 cm of the Barrett's oesophagus segment at endoscopy, and snap-frozen. A snap-frozen section was taken from each Barrett's oesophagus sample to determine the grade of dysplasia. Patients in the pre-cancer categories who received previous ablative treatment were excluded. Samples with squamous contamination were excluded.

### Cambridge sequencing data

Sequencing was performed for cases with an estimated tumour purity of higher than 70%, as determined by a pathologist. WGS by Illumina (100–150-bp paired-end reads) was performed with 50-fold coverage for the tumour and 30-fold coverage for the matched germline control. Reads were then aligned with BWA-MEM[51] to GRCh37 (1000 Genomes Project human_g1k_v37 with decoy sequences hs37d5). Aberrant cell fraction and ploidy were previously reported[13] and were generated using ASCAT v.2.3 (ref. 48).

### Detection of focal amplifications in the Cambridge cohort

Both Cambridge BAM files were aligned to GRCh37 (1000 Genomes Project human_g1k_v37 with decoy sequences hs37d5) using BWA-MEM v.0.7.17. Absolute copy number profiles were generated using ASCAT v.2.3 (ref. 48). Genomic regions with a total copy number greater than 4.5 and an interval size greater than 10 kbp were identified, merged and refined with the amplified_intervals.py script. Each seed region was given to AmpliconArchitect separately to improve runtime on each sample. AmpliconArchitect was run in the default explore mode to reconstruct amplicon structures and amplicons formed by the same regions were deduplicated on the basis of genomic overlap such that for overlapping AmpliconArchitect amplicons, the amplicon with the highest-level classification was kept (ranked by ecDNA, BFB, complex non-cyclic and then linear), with ties being broken by largest amplicon size.

### Detection of focal amplifications in the TCGA cohort

We used the Dockerized AmpliconSuite-pipeline wrapper to detect focal amplifications in the TCGA cohort. The wrapper pipeline for seed detection incorporated CNVKit v.0.9.7 (ref. 47) run in unpaired mode to detect CNVs. The CNV calls were then provided with the amplified_intervals.py script and filtered on the basis of regions having a copy number greater than 4.5 and a size larger than 50 kbp to produce a set of seed regions. We used AmpliconArchitect to infer the architecture of amplicons, The pipeline was run on 20 TCGA-ESCA EAC tumour WGS BAM files, aligned to GRCh37, through the Institute for Systems Biology Cancer Genomics Cloud (https://isb-cgc.appspot.com/), which provides a cloud-based platform for TCGA data analysis.

### FHCC sequencing data and annotations

Sequencing data for the FHCC study were previously published[14]. All research participants who contributed clinical data and biospecimens to this study provided written informed consent, subject to oversight by the Fred Hutchinson Cancer Center IRB Committee D (reg. ID 5619). All samples collected for the FHCC study were from patients who had not received treatment (treatment-naive). Reads were then aligned

with BWA-MEM (v.0.6.2-r126)[51] to GRCh37 (1000 Genomes Project human_g1k_v37 with decoy sequence hs37d5). BAM files underwent subsequent indel realignment with GATK IndelRealigner v.3.4-0-g7e26428 (ref. 52). Chromothripsis calls were derived from a previous study[22]. Genome-doubling (WGD) calls were derived from another previous study[14]. Purity and ploidy were also assessed as described previously[14], using pASCAT v.2.1.

### Detection of focal amplifications in the FHCC cohort
We used the AmpliconSuite-pipeline wrapper to detect focal amplifications in the FHCC cohort. The wrapper pipeline for seed detection incorporated CNVKit v.0.9.6 (ref. 47) run in tumour-normal mode to call somatic CNVs against the matched normal WGS samples for each patient (when multiple normal samples were available, one was selected arbitrarily). Normal samples also underwent the same pipeline in unpaired mode for stand-alone CNV detection. The CNV calls were then provided the amplified_intervals.py script and filtered on the basis of regions having a copy number greater than 4.3 (4.0 for normal samples) and size larger than 50 kbp (10 kbp for normal samples) to produce a set of seed regions. The wrapper then invoked AmpliconArchitect in default mode on the WGS BAM files to examine seed regions and profile the architecture of the focal amplifications. The resulting graph and cycles output files were provided to AmpliconClassifier v.0.4.13 to produce classifications of the AmpliconArchitect amplicons for ecDNA, BFB, complex non-cyclic and linear focal amplifications (Supplementary Information). AmpliconClassifier also specified BED files corresponding to the classified regions and annotated the identity of genes on the focal amplifications.

### FHCC cohort histology
The histology data from FHCC are a re-analysis of the previously published cohort[14]. In brief, the biopsy samples that underwent WGS were not assessed for pathological diagnosis, by design. Instead, the pathological analysis was performed from adjacent-level biopsies from the oesophagus, as described before[14]. If a sequencing biopsy had a histology biopsy from the same level along the oesophagus (measured from the gastro-oesophageal junction), then it was denoted as having on-level histology. If a sequencing biopsy had a histology biopsy from within ±1 cm of the same level, it was denoted as having windowed histology. When multiple histology samples could be paired with the sequencing, the histology biopsy with the most severe disease state was assigned.

### TP53 alteration analysis
In the FHCC cohort, TP53 status was determined from a previous study[14] and we defined TP53 alteration as cases in which either single (+/−) or double (−/−) loss of TP53 was detected. In brief, for the FHCC cohort, mutations were defined as any moderate- to high-impact SNV or indel as reported by SNPeff[53]. Deletions of at least one exon, or structural variants affecting the TP53 coding sequence or splice sites, were also considered to disrupt TP53, as were copy number alterations that affected at least half of the exonic regions. All alterations were verified manually using IGV[54] or Partek. For the Cambridge cohort, TP53 status was determined by identifying somatic coding variants (missense, frameshift, stop-gain or splice-site variants), using Strelka v.2.0.15 (ref. 55) and Variant Effect Predictor v.78 (ref. 56). Alteration was defined as one or more copies of TP53 being affected by a mutational event.

### Selection of gene lists
Oncogenes were derived from a combination of the ONGene database[57], as well as Barrett's oesophagus and EAC driver genes listed in previous reports[14,58,59]. The complete list is provided in Supplementary Table 8. Immunomodulatory genes were derived from the HisgAtlas database[60]. When evaluating the presence of genes on ecDNA, the average gene copy number was required to be 4.5 or higher and the 5′ end intact.

### Reporting summary
Further information on research design is available in the Nature Portfolio Reporting Summary linked to this article.

### Data availability
The Barrett's oesophagus and HGD Cambridge UK cohort WGS data, histology and metadata have been previously published[13], and WGS data are available through the European Genome-phenome Archive (EGA) under accession number EGAD00001006349. The EAC Cambridge UK cohort WGS data, histology and metadata were downloaded from the International Cancer Genome Consortium (ICGC; https://dcc.icgc.org/) under accession number EGAD00001002156. The FHCC cohort WGS samples, histology and metadata have been previously published[14], and WGS data are available from the NCBI dbGaP database under accession number phs001912.v1.p1. All sequencing data, histology and metadata for TCGA were downloaded from the Genomic Data Commons (GDC; https://gdc.cancer.gov/) under accession number phs000178.v11.p8. We have uploaded the AmpliconArchitect and AmpliconClassifier output files to figshare at https://doi.org/10.6084/m9.figshare.21893826.

### Code availability
We developed and used the following publicly available codebases: AmpliconArchitect (https://github.com/jluebeck/AmpliconArchitect), AmpliconClassifier (https://github.com/jluebeck/AmpliconClassifier), AmpliconSuite-pipeline (https://github.com/jluebeck/AmpliconSuite-pipeline) and CycleViz (https://github.com/jluebeck/CycleViz).

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

**Acknowledgements** We thank the many individuals who donated their tissue samples to scientific research, without whom this study could not have taken place. This work was delivered as part of the eDyNamiC team supported by the Cancer Grand Challenges partnership funded by Cancer Research UK (CRUK) (P.S.M. and H.Y.C., CGCATF-2021/100012; V.B. and J.L., CGCATF-2021/100025; S.W., CGCATF-2021/100023; and R.G.W.V., CGCATF-2021/100016) and the National Cancer Institute (P.S.M. and H.Y.C., OT2CA278688; V.B. and J.L., OT2CA278635; S.W., OT2CA278683; and R.G.W.V., OT2CA278649). P.S.M. is the eDyNamiC team lead and co-authors R.G.W.V., H.Y.C., S.W. and V.B. are members of the eDyNamiC team. This study was also supported by a grant from the National Brain Tumour Society (P.S.M.) and National Institutes of Health (NIH) R01-CA238379 (P.S.M.). V.B. and J.L. were supported in part by grants U24CA264379 and R01GM114362 from the NIH. S.W. is a scholar of and is supported by the Cancer Prevention and Research Institute of Texas (RR210034). The Cambridge research (R.C.F., A.W.T.N. and S.J.) was funded by a programme grant from the Medical Research

Council and from CRUK. The work also received infrastructure support from the NIHR Biomedical Research Centre and the CRUK Experimental Cancer Research Centre. A.C.K.-S. was funded by CRUK through a clinical research fellowship. Research at FHCC (T.G.P., B.J.R., C.A.S., P.C.G. and X.L.) was funded by NIH grants P01 CA91955 and P30 CA015704. H.K. is supported by the Brain Korea 21 Four Project, a Korean Ministry of Food and Drug Safety grant (21153MFDS607) and Korean Ministry of Science and ICT grants (NRF-2019R1A5A2027340 and NRF-2022M3C1A309202211). R.G.W.V. acknowledges support from NIH–NCI grants R01 CA237208, R21 CA256575 and R33 CA236681 and a Cancer Center Support grant P30 CA034196, as well as NIH–NINDS grant R21 NS114873. Work in the L.B.A. laboratory (L.B.A. and Y.H.) was supported by NIH grants R01ES030993-01A1 and R01ES032547-01 and a Cancer Grand Challenges Mutographs team award funded by CRUK (C98/A24032). L.B.A. is also personally supported by a Packard Fellowship for Science and Engineering. This work was also supported by the PreCancer Genome Atlas (PCGA) project with core funding from UC San Diego NCI P30 (P30 CA023100; S.M.L.), and in part by the SU2C-AACR-DT25-17 and US NIH grants R01DE026644, P30 CA023100, HHSN261201200031I and UG1CA242596. We also thank G. Devonshire, who ran the data-analysis pipeline for the Cambridge sequencing data. The oesophagus illustration shown in our study was created by scientific illustrator Tami Tolpa (Tolpa Studios, https://www.tolpa.com/).

**Author contributions** J.L., R.C.F., T.G.P., P.C.G., S.W., V.B. and P.S.M. designed and conceived the analyses. R.C.F. led the Cambridge University UK team and T.G.P. led the FHCC team. The Cambridge data were analysed by J.L., A.W.T.N., A.C.K.-S., S.J. and R.C.F. The FHCC data were analysed by J.L., P.C.G., X.L., C.A.S., Y.H., L.B.A., C.C.M., B.J.R. and T.G.P. The TCGA data were analysed by J.L., H.K. and R.G.W.V. Computational methods introduced in the manuscript were conceived by J.L. and V.B. J.L., S.W., V.B. and P.S.M. wrote the manuscript, and R.C.F., A.W.T.N, T.G.P., P.C.G. and H.Y.C. provided input during the writing of the paper. All authors provided feedback on the analyses, edited and approved the manuscript.

**Competing interests** P.S.M. is a co-founder, chairs the scientific advisory board (SAB) of and has equity interest in Boundless Bio. P.S.M. is also an advisor with equity for Asteroid Therapeutics and is an advisor to Sage Therapeutics. V.B. is a co-founder, consultant, SAB member and has equity interest in Boundless Bio and Abterra, and the terms of this arrangement have been reviewed and approved by the University of California, San Diego in accordance with its conflict of interest policies. S.W. is a member of the SAB of Dimension Genomics. H.Y.C. is a co-founder of Accent Therapeutics, Boundless Bio, Cartography Bio, and Orbital Therapeutics, and is an advisor to 10X Genomics, Arsenal Biosciences, Chroma Medicine, and Spring Discovery. R.C.F. is named on patents related to Cytosponge and associated assays that were licensed to Covidien (now Medtronic). R.C.F. has founder shares (<3%) in Cyted. R.G.W.V. is a co-founder of Boundless Bio and an advisor to Stellanova Therapeutics and NeuroTrials. L.B.A. is a compensated consultant and has equity interest in io9. His spouse is an employee of Biotheranostics. L.B.A. is an inventor on the US patent 10,776,718 for source identification by non-negative matrix factorization. L.B.A. also declares provisional patent applications for 'Clustered mutations for the treatment of cancer' (US provisional application serial number 63/289,601) and 'Artificial intelligence architecture for predicting cancer biomarker' (serial number 63/269,033). S.M.L. is a co-founder of io9, and also declares a provisional patent application for 'Methods and biomarkers in cancer' (US provisional application serial number 114198-1160). J.L. is a part-time paid consultant for Boundless Bio, and the terms of this arrangement have been reviewed and approved by the University of California, San Diego in accordance with its conflict of interest policies. P.S.M., V.B., J.L. and S.W. declare a patent application related to this work: 'Methods and compositions for detecting ecDNA' (US patent application number 17/746,748). The remaining authors declare no competing interests.

## Additional information

**Correspondence and requests for materials** should be addressed to Rebecca C. Fitzgerald, Thomas G. Paulson, Howard Y. Chang, Sihan Wu, Vineet Bafna or Paul S. Mischel.

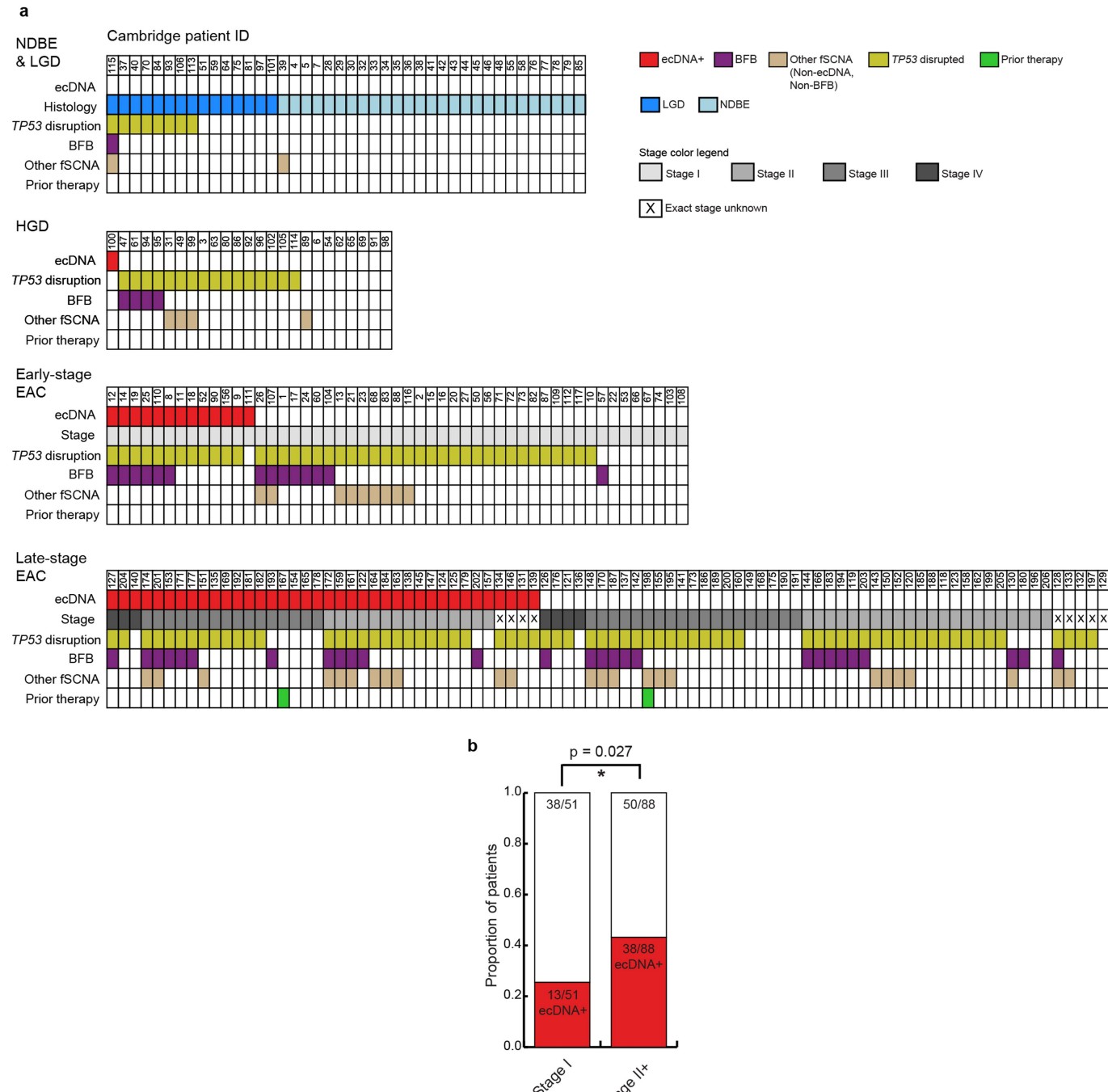

**Extended Data Fig. 1 | Oncoprint tables for the Cambridge data. a**, Oncoprint table for samples from Cambridge patient with Barrett's oesophagus and EAC segregated by histology type showing ecDNA status, histology or cancer stage (if applicable) *TP53* alteration (by mutational analysis, involving at least one copy), BFB status, other fsCNA (non-BFB, non-ecDNA) status and prior therapy (chemotherapy or radiation) on the tumours in patients with cancer. **b**, Proportion of Cambridge EAC tumour samples with ecDNA separated by tumour stage I versus stage II or higher.

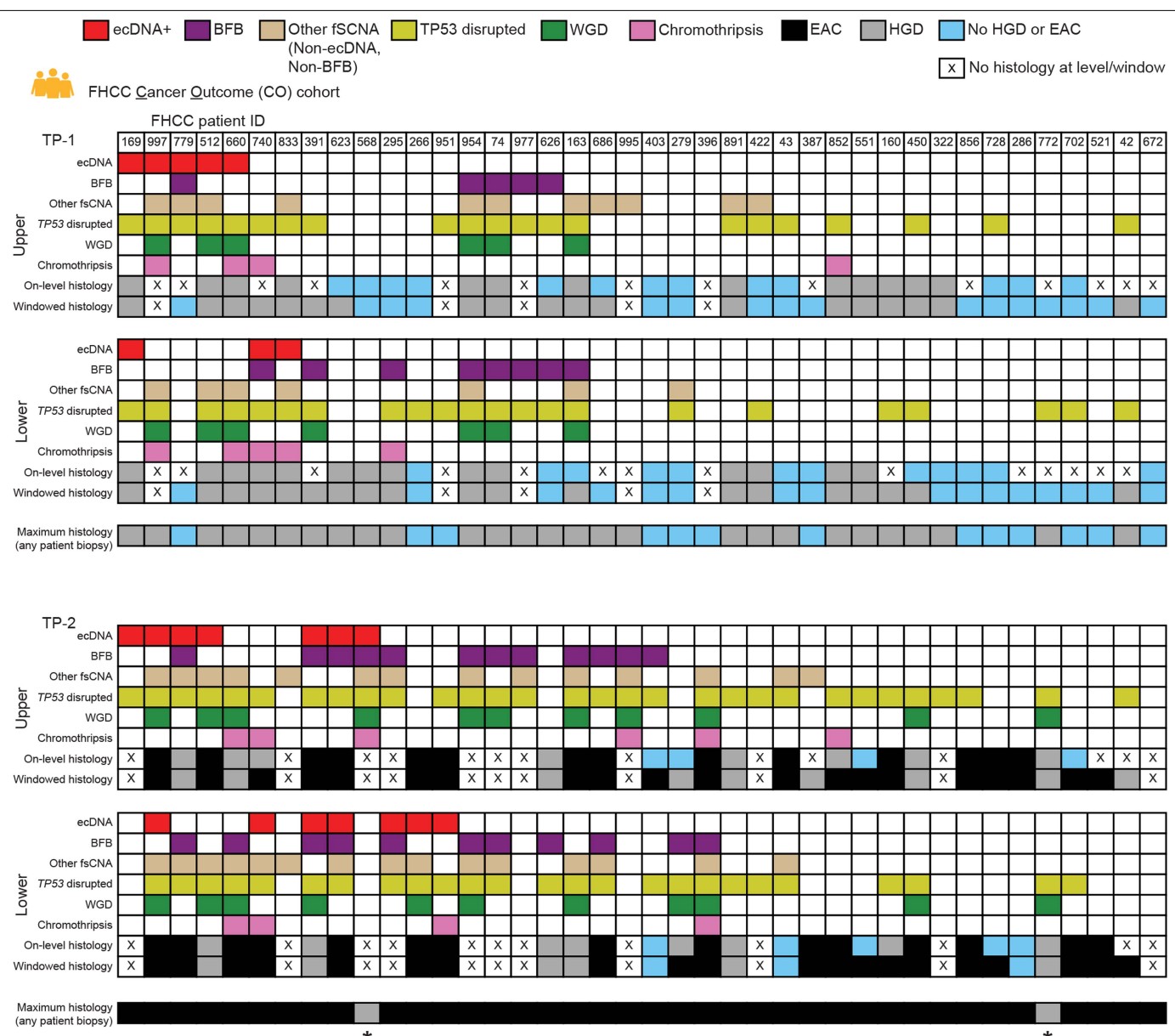

**Extended Data Fig. 2 | Oncoprint tables for the FHCC cancer-outcome data.** Oncoprint tables of samples from FHCC CO patient WGS samples encoding ecDNA status, BFB status, other fSCNA (non-BFB, non-ecDNA) status, *TP53* alteration (at least one gene copy affected), WGD status and chromothripsis status, as well as on-level and windowed histology for each time point and both upper and lower oesophageal samples for time points TP-1 and TP-2. Maximum histology from any histology biopsy is shown at the bottom of each time point. Asterisk indicates cancer diagnosis made at next endoscopy since biopsies from the diagnostic EAC endoscopy were unavailable for CO patient ID 772 and lacked sufficient DNA for CO patient ID 568, so biopsies from the penultimate endoscopy were substituted (occurring 1.44 and 8.16 months after TP-2 for patients 568 and 772, respectively).

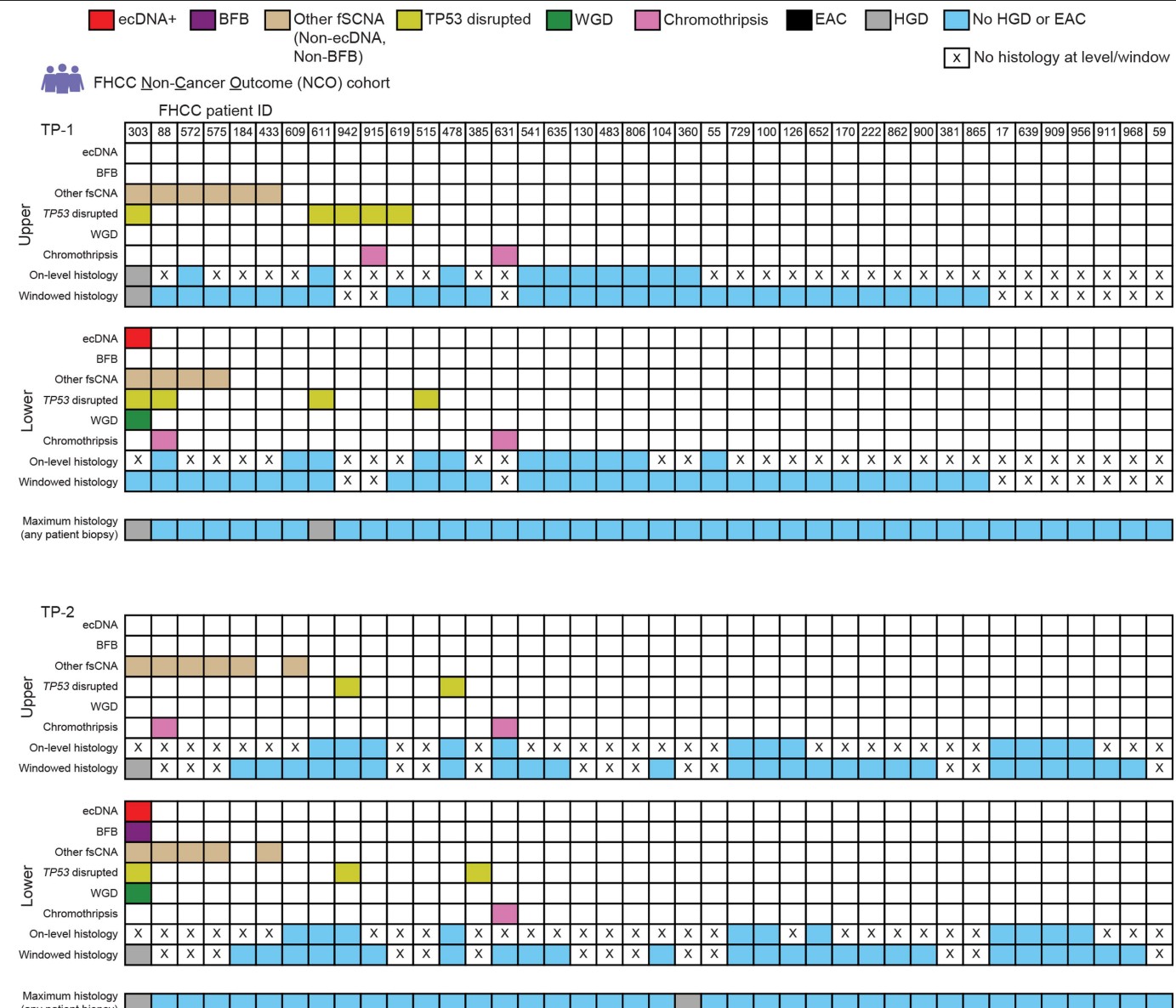

**Extended Data Fig. 3 | Oncoprint tables for the FHCC non-cancer-outcome data.** Oncoprint tables of FHCC NCO patient WGS samples encoding ecDNA status, BFB status, other fSCNA (non-BFB, non-ecDNA) status, *TP53* alteration (at least one gene copy affected), WGD status and chromothripsis status, as well as on-level and windowed histology for each time point and both upper and lower oesophageal samples for time points TP-1 and TP-2. Maximum histology from any histology biopsy is shown at the bottom of each time point.

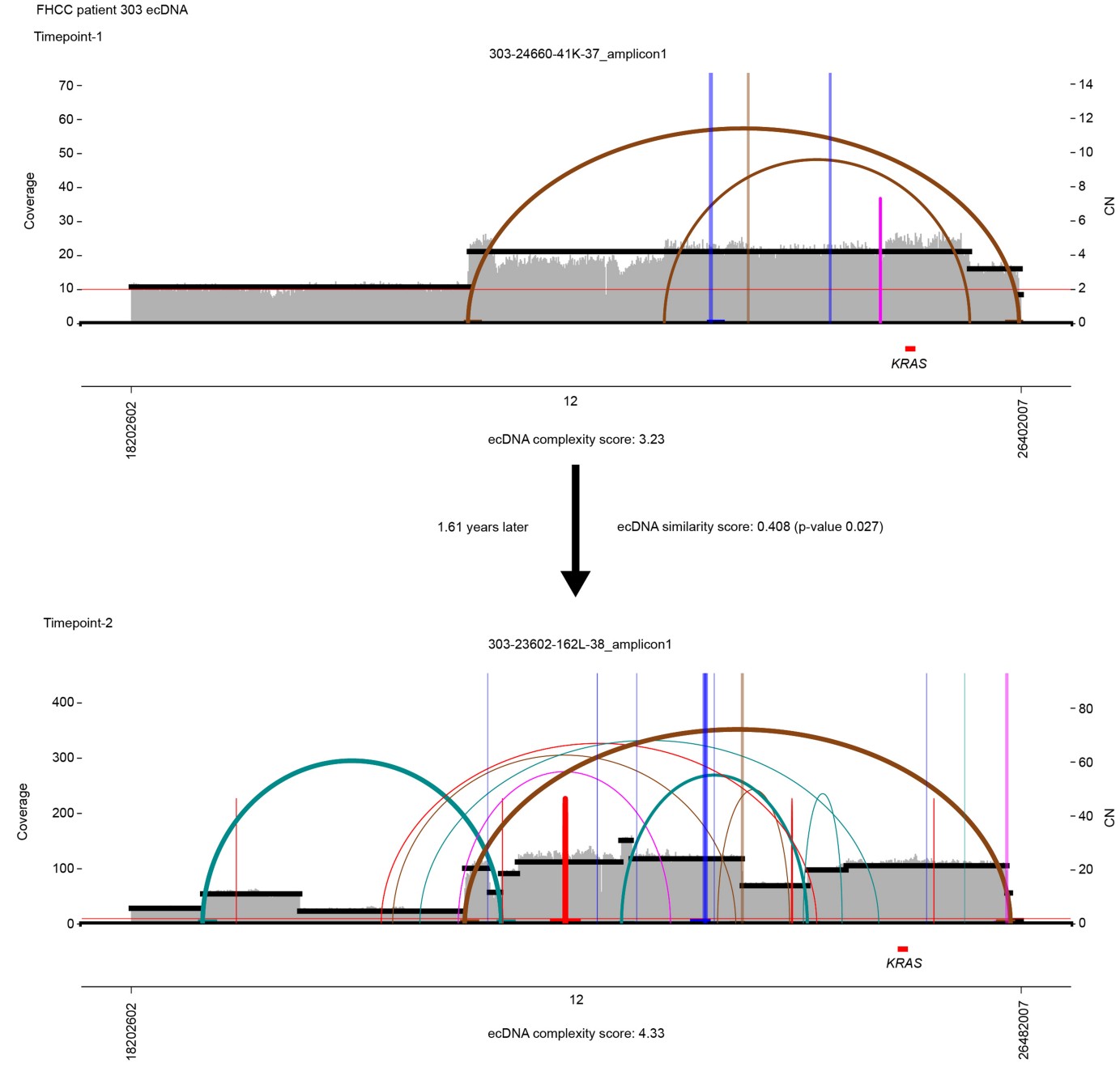

FHCC patient 303 ecDNA

Timepoint-1

303-24660-41K-37_amplicon1

ecDNA complexity score: 3.23

1.61 years later    ecDNA similarity score: 0.408 (p-value 0.027)

Timepoint-2

303-23602-162L-38_amplicon1

ecDNA complexity score: 4.33

**Extended Data Fig. 4 | Analysis of ecDNA evolution.** The *KRAS*-bearing ecDNA focal amplification detected in biopsies from FHCC NCO patient 303 at time point TP-1 and time point TP-2. Amplicon similarity analysis reveals a common origin of the ecDNA, and ecDNA copy number and complexity increased during the 1.61 years between samples. *P* value assessed against a beta-distribution model fit to distribution of similarity scores among genomically overlapping focal amplifications from independent samples (Supplementary Information).

# a

■ ecDNA+　■ BFB　■ Other fsCNA (Non-ecDNA, Non-BFB)　■ TP53 disrupted　■ WGD　■ Chromothripsis　■ EAC　■ HGD　■ No HGD or EAC

[x] Data unavailable　■ Patient included in long-term follow-up

FHCC NCO cohort long-term follow-ups

FHCC patient ID

| | 303 | 88 | 572 | 575 | 184 | 433 | 609 | 611 | 942 | 915 | 619 | 515 | 478 | 385 | 631 | 541 | 635 | 130 | 483 | 806 | 104 | 360 | 55 | 729 | 100 | 126 | 652 | 170 | 222 | 862 | 900 | 381 | 865 | 17 | 639 | 909 | 956 | 911 | 968 | 59 |

**Upper**

ecDNA, BFB, Other fsCNA, TP53 disrupted, WGD, Chromothripsis, On-level histology, Windowed histology

**Lower**

ecDNA, BFB, Other fsCNA, TP53 disrupted, WGD, Chromothripsis, On-level histology, Windowed histology

Maximum histology (any patient biopsy)

# b

FHCC NCO time from TP-2 to last known vital status

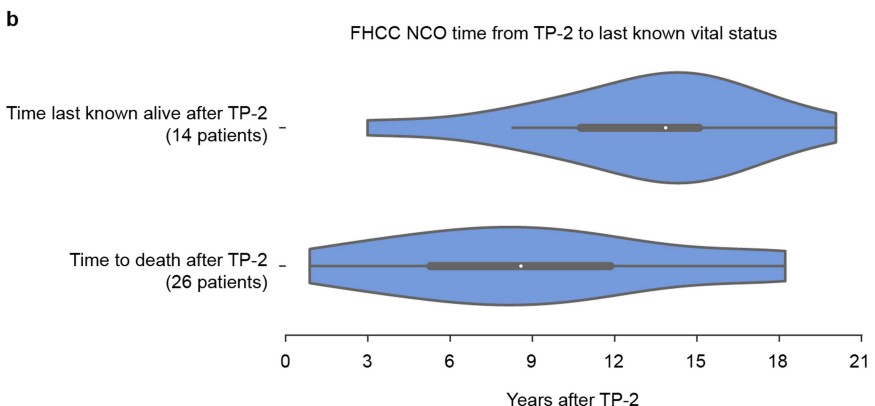

Time last known alive after TP-2 (14 patients)

Time to death after TP-2 (26 patients)

Years after TP-2

**Extended Data Fig. 5 | Oncoprint tables for FHCC non-cancer-outcome long-term follow-ups. a**, Oncoprint tables of biopsies from NCO patients from the FHCC cohort with long-term follow-ups (in orange, collected median 9.6 years after TP-2). **b**, Distribution of FHCC NCO follow-up durations from TP-2 to the time at which the patient was last known to be alive (top, mean = 13.9 years) or TP-2 to death (bottom, mean = 8.6 years).

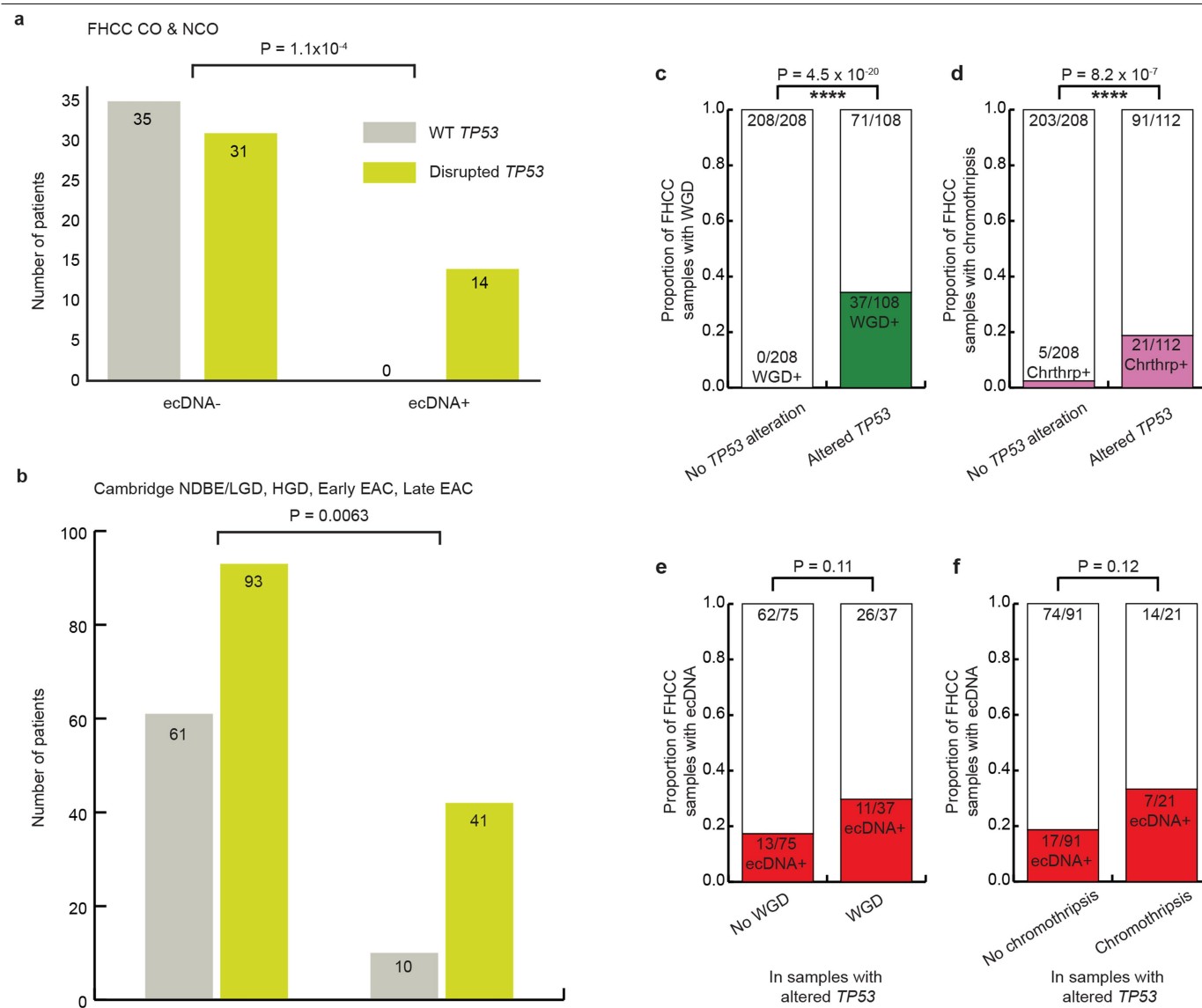

**Extended Data Fig. 6 | Association of ecDNA with other genomic features.**
**a**, Association of ecDNA presence and *TP53* status in biopsies from patients from the FHCC cohort. **b**, Association of ecDNA presence and *TP53* status in samples from patients from the Cambridge cohort, respectively for FHCC and Cambridge. **c**, Proportion of FHCC samples with WGD separated by *TP53* alteration status. **d**, Proportion of FHCC samples with chromothripsis separated by *TP53* alteration status. **e**, Proportion of *TP53* alteration FHCC samples with ecDNA, separated by WGD status. **f**, Proportion of *TP53* alteration FHCC samples with ecDNA, separated by chromothripsis status. All statistical differences in frequencies were assessed by one-sided Fisher's exact test.

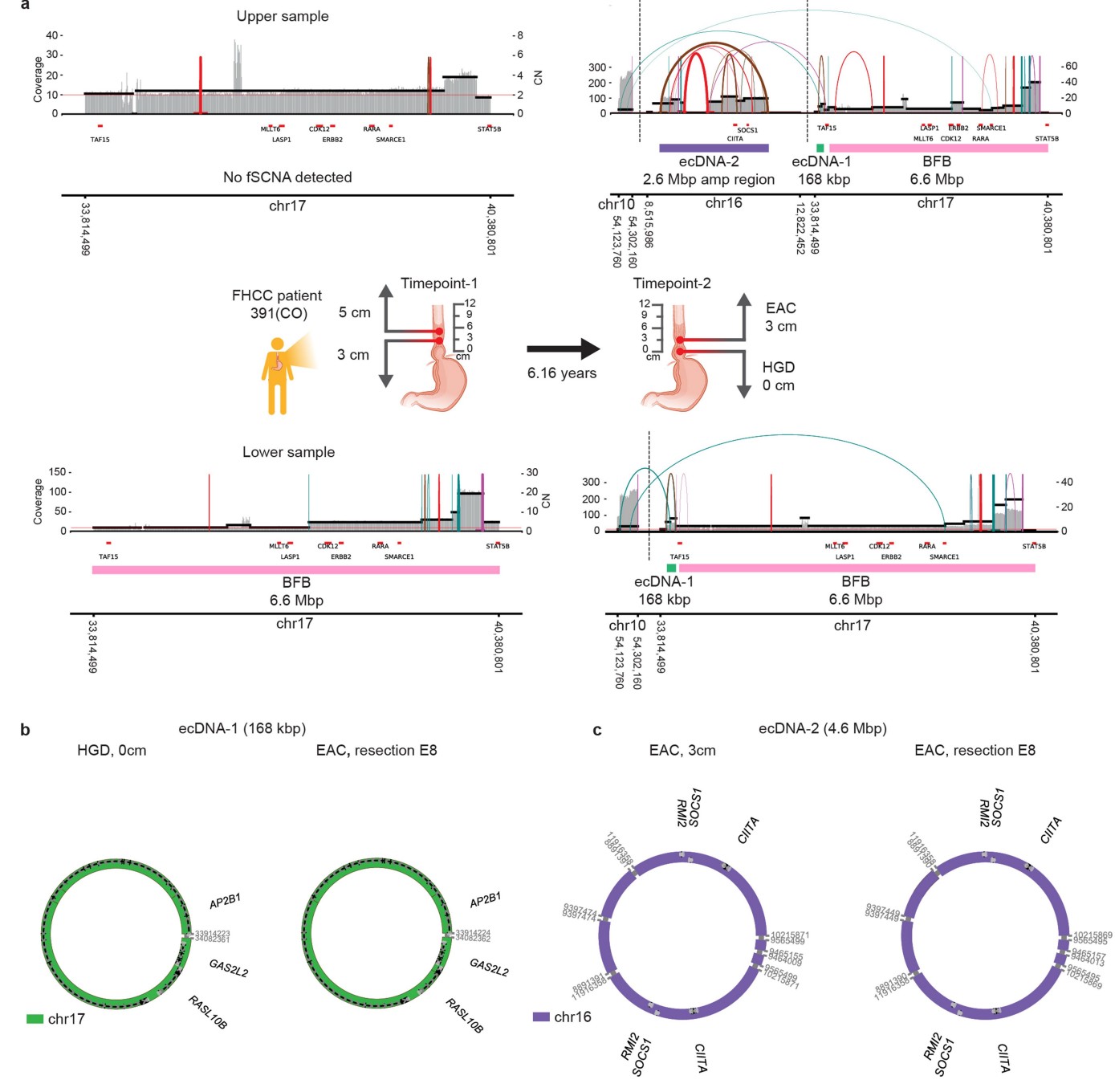

**Extended Data Fig. 7 | Focal amplification evolution. a**, Barrett's oesophagus segment samples from patient 391 show conserved focal amplification of BFB and emergence of ecDNA between time points TP-1 and TP-2. **b**, The structure of ecDNA-1 detected in the lower pre-cancer sample from TP-2 in patient 391, in which HGD was in the histology window, and an identical structure derived from the adenocarcinoma resection. **c**, The structure of ecDNA-2, detected in the upper sample from TP-2 in patient 391, in which EAC was present in the histology window, and an identical structure derived from the adenocarcinoma resection. Amplicon similarity analysis of ecDNA-1 and -2 reveals common origins of the structures.

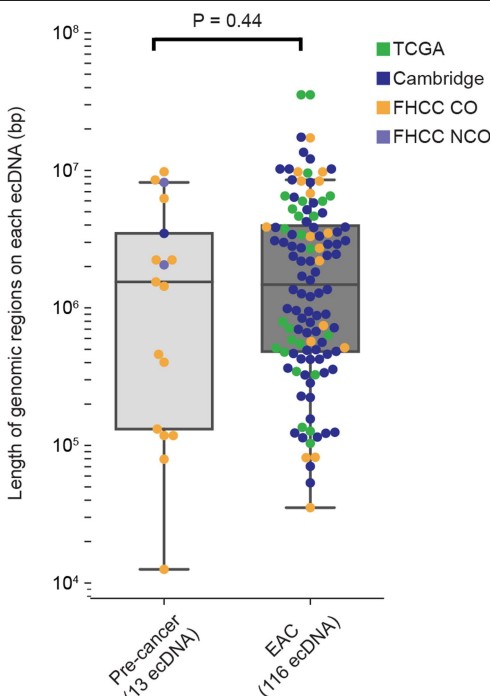

**Extended Data Fig. 8 | Sizes of intervals captured on ecDNA.** The length of predicted genomic intervals captured on ecDNA, visualized on a $\log_{10}$ scale, for each distinct ecDNA in the combined cohorts, compared by pre-cancer versus EAC (Mann–Whitney $U$ test, two-sided).

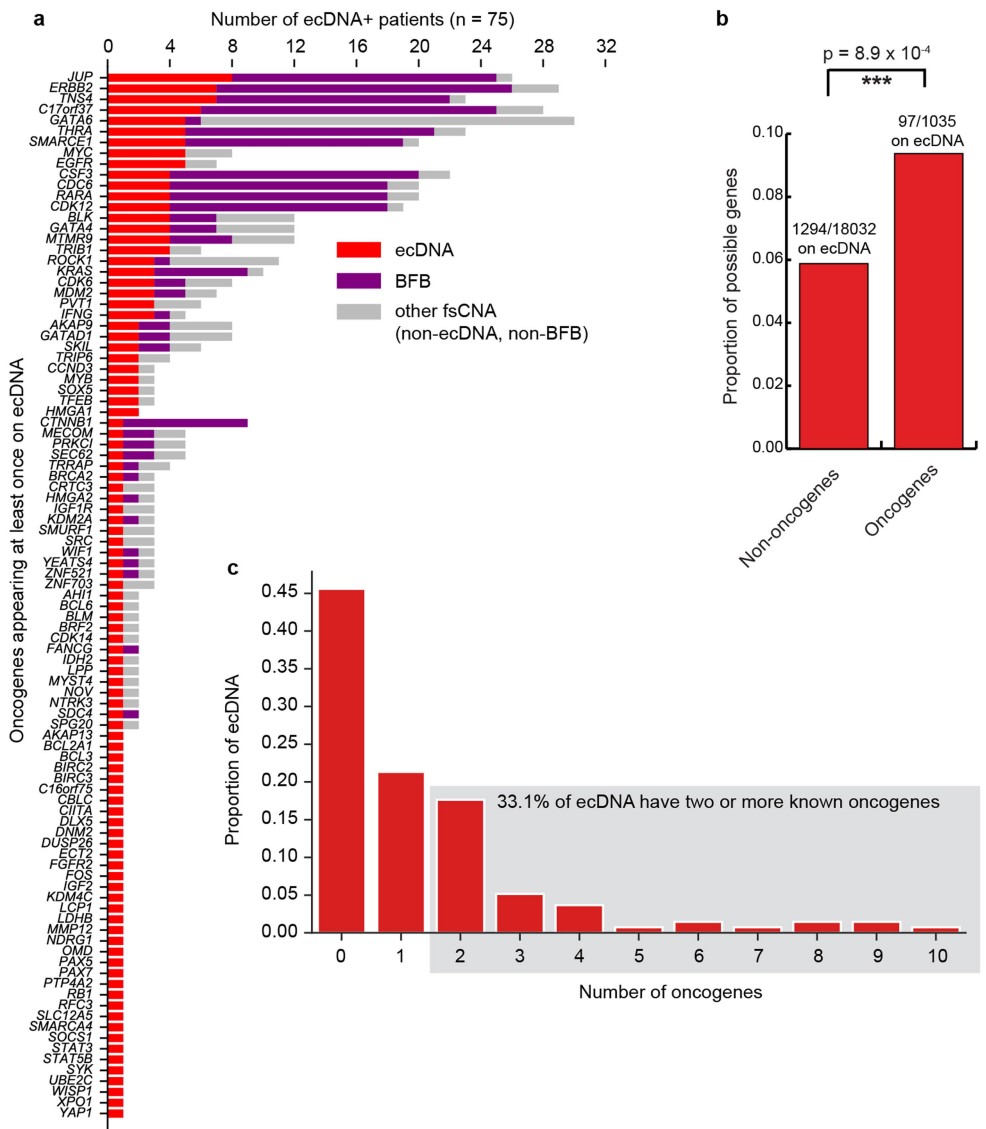

**Extended Data Fig. 9 | Frequencies of ecDNA-borne oncogenes. a**, For oncogenes detected on ecDNA in samples from at least one patient, the number of patients with at least one sample having the oncogene listed on ecDNA, and the frequency of that gene on other types of focal amplifications. **b**, Proportion of the set of possible unique genes on ecDNA, separated by oncogene status. Difference assessed by one-sided Fisher's exact test. **c**, Distribution of the number of oncogenes on individual ecDNA.

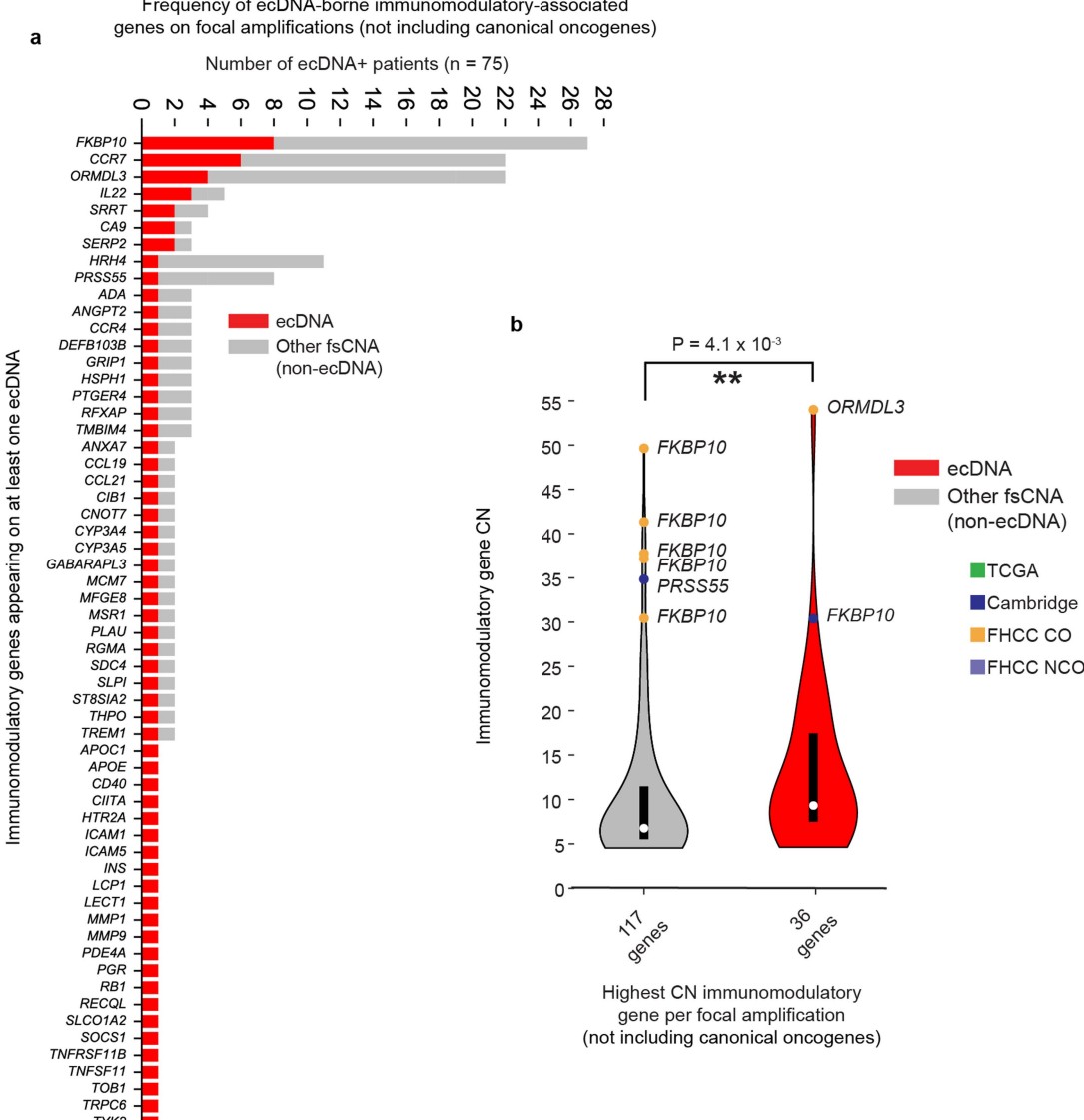

**Extended Data Fig. 10 | Frequencies of ecDNA-borne immunomodulatory genes. a**, For immunomodulatory-associated genes detected on ecDNA, the number of patients with at least one sample having the gene listed on ecDNA, and the frequency of that gene on other focal amplifications as well. **b**, Copy number for the highest copy number focally amplified immunomodulatory-associated gene in each unique amplicon that was ecDNA or non-ecDNA fsCNA. ecDNAs show a significantly higher copy number of immunomodulatory-associated genes on ecDNA versus non-ecDNA fsCNA (Mann–Whitney $U$ test, one-sided).

# Reporting Summary

## Statistics

For all statistical analyses, confirm that the following items are present in the figure legend, table legend, main text, or Methods section.

| n/a | Confirmed | |
|---|---|---|
| ☐ | ☒ | The exact sample size (*n*) for each experimental group/condition, given as a discrete number and unit of measurement |
| ☐ | ☒ | A statement on whether measurements were taken from distinct samples or whether the same sample was measured repeatedly |
| ☐ | ☒ | The statistical test(s) used AND whether they are one- or two-sided *Only common tests should be described solely by name; describe more complex techniques in the Methods section.* |
| ☐ | ☒ | A description of all covariates tested |
| ☒ | ☐ | A description of any assumptions or corrections, such as tests of normality and adjustment for multiple comparisons |
| ☐ | ☒ | A full description of the statistical parameters including central tendency (e.g. means) or other basic estimates (e.g. regression coefficient) AND variation (e.g. standard deviation) or associated estimates of uncertainty (e.g. confidence intervals) |
| ☐ | ☒ | For null hypothesis testing, the test statistic (e.g. *F*, *t*, *r*) with confidence intervals, effect sizes, degrees of freedom and *P* value noted *Give P values as exact values whenever suitable.* |
| ☒ | ☐ | For Bayesian analysis, information on the choice of priors and Markov chain Monte Carlo settings |
| ☒ | ☐ | For hierarchical and complex designs, identification of the appropriate level for tests and full reporting of outcomes |
| ☒ | ☐ | Estimates of effect sizes (e.g. Cohen's *d*, Pearson's *r*), indicating how they were calculated |

*Our web collection on statistics for biologists contains articles on many of the points above.*

## Software and code

Policy information about availability of computer code

| | |
|---|---|
| Data collection | No data were generated for this study. All data can be downloaded from the included repositories, links and references. The BE and HGD Cambridge UK cohort whole-genome sequencing data, histology and metadata was previously published in Katz-Summercorn et al.13 and whole-genome sequencing data are available through the European Genome-phenome Archive (EGA) under accession ID EGAD00001006349. The EAC Cambridge UK cohort whole-genome sequencing data, histology and metadata were downloaded from the International Cancer Genome Consortium (ICGC) at https://dcc.icgc.org/. The FHCC cohort whole genome sequencing samples, histology and metadata were previously published in Paulson et al.14 and whole-genome sequencing data are available from the NCBI dbGaP database under accession ID phs001912.v1.p1. All sequencing data, histology and metadata for TCGA were downloaded from the GDC (https://gdc.cancer.gov/) under accession ID phs000178.v11.p8. We have uploaded the AmpliconArchitect and AmpliconClassifier output files to FigShare at https://doi.org/10.6084/m9.figshare.21893826. |
| Data analysis | AmpliconSuite-pipeline (0.1203.12) https://github.com/jluebeck/AmpliconSuite-pipeline<br>AmpliconArchitect (1.2) (https://github.com/jluebeck/AmpliconArchitect)<br>AmpliconClassifier (0.4.13) (https://github.com/jluebeck/AmpliconClassifier)<br>CycleViz (0.1.1) (https://github.com/jluebeck/CycleViz)<br>SciPy (1.9.1)<br>ASCAT (v2.3)<br>CNVKit (version 0.9.7 and version 0.9.6)<br>ISTAT (1.0.0)<br>BWA-mem (version 0.7.17 and version 0.6.2-r126)<br>GATK IndelRealigner (3.4-0-g7e26428)<br>SNPeff (4.2) |

Strelka (2.0.15)
Variant Effect Predictor (78)

For manuscripts utilizing custom algorithms or software that are central to the research but not yet described in published literature, software must be made available to editors and reviewers. We strongly encourage code deposition in a community repository (e.g. GitHub). See the Nature Portfolio guidelines for submitting code & software for further information.

# Data

Policy information about availability of data

All manuscripts must include a data availability statement. This statement should provide the following information, where applicable:
- Accession codes, unique identifiers, or web links for publicly available datasets
- A description of any restrictions on data availability
- For clinical datasets or third party data, please ensure that the statement adheres to our policy

The BE and HGD Cambridge UK cohort whole-genome sequencing data, histology and metadata was previously published in Katz-Summercorn et al.13 and whole-genome sequencing data are available through the European Genome-phenome Archive (EGA) under accession ID EGAD00001006349. The EAC Cambridge UK cohort whole-genome sequencing data, histology and metadata were downloaded from the International Cancer Genome Consortium (ICGC) at https://dcc.icgc.org/. The FHCC cohort whole genome sequencing samples, histology and metadata were previously published in Paulson et al.14 and whole-genome sequencing data are available from the NCBI dbGaP database under accession ID phs001912.v1.p1. All sequencing data, histology and metadata for TCGA were downloaded from the GDC (https://gdc.cancer.gov/) under accession ID phs000178.v11.p8. We have uploaded the AmpliconArchitect and AmpliconClassifier output files to FigShare at https://doi.org/10.6084/m9.figshare.21893826.

# Human research participants

Policy information about studies involving human research participants and Sex and Gender in Research.

| | |
|---|---|
| Reporting on sex and gender | Sex and gender data was not collected in this study, as we reanalyzed previously published collections of samples from human research participants. In all study sources, sex and gender were previously reported and analyzed. |
| Population characteristics | Population characteristics for each study used as a source of data are described in<br>FHCC cohort: Paulson et al., Nature Communications 2022<br>Cambridge Cohort: Katz-Summercorn et al., Nature Communications 2022 and The ICGC/TCGA Pan-Cancer Analysis of Whole Genomes Consortium, Nature 2020.<br>TCGA: The Cancer Genome Atlas Research Network, Nature 2017 |
| Recruitment | Recruitment for each study used as a source of data are described in<br>FHCC cohort: Paulson et al., Nature Communications 2022<br>Cambridge Cohort: Katz-Summercorn et al., Nature Communications 2022 and The ICGC/TCGA Pan-Cancer Analysis of Whole Genomes Consortium, Nature 2020.<br>TCGA: The Cancer Genome Atlas Research Network, Nature 2017 |
| Ethics oversight | Ethics oversight for each study used as a source of data are described in<br>FHCC cohort: Paulson et al., Nature Communications 2022<br>Cambridge Cohort: Katz-Summercorn et al., Nature Communications 2022 and The ICGC/TCGA Pan-Cancer Analysis of Whole Genomes Consortium, Nature 2020.<br>TCGA: The Cancer Genome Atlas Research Network, Nature 2017 |

Note that full information on the approval of the study protocol must also be provided in the manuscript.

# Field-specific reporting

Please select the one below that is the best fit for your research. If you are not sure, read the appropriate sections before making your selection.

☒ Life sciences　　　☐ Behavioural & social sciences　　　☐ Ecological, evolutionary & environmental sciences

For a reference copy of the document with all sections, see nature.com/documents/nr-reporting-summary-flat.pdf

# Life sciences study design

All studies must disclose on these points even when the disclosure is negative.

| | |
|---|---|
| Sample size | For the Cambridge UK cohort, no sample size calculation was performed. The sample size was made as large as it could be possible based on the availability of suitable material for sequencing. For the Fred Hutch cohort, the 80 patients with Barrett's esophagus (40 with cancer outcome and 40 without cancer outcome) were selected from a previously published case-cohort study of 248 patients from Li et al, Cancer Prev Res, 2014 (PMID 24253313), in which somatic chromosomal alterations (SCA) had been characterized every two centimeters (cm) in the Barrett's segment. All samples used in this research were published in DOI 10.1038/s41467-022-29767-7 and 10.1038/s41467-022-28237-4 |
| Data exclusions | No data was excluded as it is a re-analysis of previous published whole-genome sequencing data. |

| | |
|---|---|
| Replication | This is a re-analysis of whole-genome sequencing data. Replication is not applicable. |
| Randomization | For the Cambridge UK cohort, randomization is not applicable, because samples were grouped based on the grade of pathology of the patients. For the Fred Hutch cohort, for each cancer outcome case, non-cancer outcome controls were randomly matched on baseline total SCA, age at T1, time between T1 and T2, and gender for the comparisons of the two groups. |
| Blinding | All individuals performing whole-genome sequencing analysis were blinded to cancer outcome. |

# Reporting for specific materials, systems and methods

We require information from authors about some types of materials, experimental systems and methods used in many studies. Here, indicate whether each material, system or method listed is relevant to your study. If you are not sure if a list item applies to your research, read the appropriate section before selecting a response.

## Materials & experimental systems

| n/a | Involved in the study |
|---|---|
| ☒ ☐ | Antibodies |
| ☒ ☐ | Eukaryotic cell lines |
| ☒ ☐ | Palaeontology and archaeology |
| ☒ ☐ | Animals and other organisms |
| ☒ ☐ | Clinical data |
| ☒ ☐ | Dual use research of concern |

## Methods

| n/a | Involved in the study |
|---|---|
| ☒ ☐ | ChIP-seq |
| ☒ ☐ | Flow cytometry |
| ☒ ☐ | MRI-based neuroimaging |

