## [Peer Review File · Nature]

Manuscript Title: Extrachromosomal DNA in the cancerous transformation of Barrett's esophagus

Redactions – unpublished data

Reviewer Comments & Author Rebuttals

Reviewer Reports on the Initial Version:

Referees' comments:

Referee #1 (Remarks to the Author):

Jens Luebeck and colleagues have presented an exhaustive and comprehensive analysis of genomic datasets from 3 separate cohorts of Barrett's Esophagus patients from an extrachromosomal DNA perspective. These three cohorts are well-characterized and provide complimentary, but diverse, patient populations with Barrett's esophagus and include: a cross sectional cohort of BE from Cambridge with both non-dysplastic (NDBE) and neoplastic (HGD/EAC) Barrett's esophagus; a longitudinal cohort from Seattle with NDBE patients whom either progressed (termed in this manuscript "CO") or did not progress (termed NCO) to neoplastic BE; finally an additional validation cohort was obtained from a subset of EAC patients in the TCGA with either early or advanced invasive EAC. Using a WGS analytical pipeline developed by members of authorship (AmpliconArchitect), Luebeck and colleagues have been able to characterize the pattern, spatial representation, and timing of ecDNA in relation to histological progression of Barrett's esophagus from NDBE to neoplastic lesions (HGD/EAC). Building on recent literature from the Cambridge group showing early variant amplification preceding histological progression, the authors of this present submission demonstrate that this finding may be in fact be elaborated by oncogenic and immunomodulatory gene representation on ecDNA. The presence of ecDNA correlated with disease progression, and with stage of disease (extrapolated by "early" and "late").

The authors have submitted an important study significantly advancing our understanding of the genomic drivers of Barrett's esophagus and how variant amplifications may be elaborated through ecDNA mechanisms. This work is important and though the results have identified a novel biomarker for risk stratification or a novel target for intervention.

The selected cohorts are well established and appropriate to address the research question. The methodological framework is sound as are the computational analyses. Although the methodology is sound, I have several comments and questions on the selected cohorts that may help improve this otherwise well written manuscript:

- 1) In the Abstract and Discussion, the latter which is rather brief, it would help the reader to set the frame. What is the clinical significance of ecDNA - how will this impact patient care if true - can ecDNA be targeted directly independent of the amplified gene. Please set the clinical context and significant of this work.
- 2) In the Study Samples and Methodology sections please describe these cohorts in more detail. e.g. What is the minimal duration of the follow up in the non-progressor cohort who did not develop HGD/EAC in the cross sectional cohort(Cambridge)? Why was LGD included in the non-progressor group - a diagnosis with a high degree of pathology disagreement but with a clearly defined risk of progression justifying invasive intervention (e.g ablation). In this context can you describe the review

process that confirmed the pathologic diagnosis? These two (Cambridge and FHCC) cohorts are well characterized, I'm certain that this information is available, or a prior publication can be cited. For the TP-1 and TP-2 in the patient who did NOT develop cancer, what is the minimal time interval.

3) Although there is a school of thought professing that all EAC originates from BE, this remains controversial. Do the authors have the granularity of clinical annotated data for the TCGA cohort (doubtful) to determine if the selected cohort were associated with BE? Or can they glean additional markers of BE from the sequencing data (e.g. TFF3) to confirm the origin of the EAC.

4) For the Cambridge and FHCC cohorts, Do the authors have the percent of each histological diagnosis in the FFPE sections prior to sequencing in the Cambridge diagnosis? Was clinical pathological followed (i.e. the highest grade determines diagnosis even if 1% of the sample) or a more strict definition applied to have a more accurate representation of the tissue analyzed by WGS? Would the authors clarify what they are trying to convey on line 306-307, do they mean that for adenocarcinoma a threshold of 70% of the epithelial cells have to be malignant to be included for sequencing? That appears rather strict. What percent of the tissue is "contaminated" by lower grade epithelia – could a supplemental table be included?

5) As alluded to above, the discussion is rather brief and does not put important results in context with the larger clinical picture and unmet clinical needs for Barrett's esophagus. Please expand the discussion and include a brief clinical implication of the work. How can early detection of ecDNA in non-dysplastic BE influence the management of patients? Risk stratification for shorter/longer surveillance intervals? early invasive intervention (e.g. ablation), targeting ecDNA directly to prevent/revert/reprogram "damned" mucosa?

Referee #2 (Remarks to the Author):

This manuscript by Luebeck et al is the first to examine ecDNA frequency and contribution to pathogenesis in the Barrett metaplasia, dysplasia, adenocarcinoma progression sequence. Using two separate large cohorts, the authors showed that ecDNA frequency increased from metaplasia to early and late stage cancer using WGS and Amplicon analysis. In longitudinal samples from a single patient, acquisition of a second ecDNA was associated with progression to adenocarcinoma from HGD associated with an ecDNA. Finally analysis of ecDNA in cancer compared to pre-cancer lesions demonstrated that cancer ecDNA were more complex and coded for oncogenes not seen on non-ecDNA amplifications and immunomodulatory genes. In one cohort that followed patients over time, the presence of ecDNA predicted progression to EAC.

Major comments:

1) Since ecDNA generation could occur due to therapeutic interventions (e.g. ablation, chemotherapy, radiation) and resulting DNA damage, especially in more advanced disease, further description of the patient cohorts would be useful. The Cambridge cohort of 42 non-dysplastic BE and 25 HGD was described as a surveillance cohort in the text, but in Figure 1 the HGD patients were described as having received therapy. No description of treatment was provided for the FHCRC cohort, were these patients with HGD followed without intervention? What was the length of follow-up of the Cambridge NDBE cohort and the FHCRC NCO cohorts (9.6 years was provided for only a subset)?

- 2) How accurate are AmpliconArchitect and AmpliconClassifier at identifying ecDNA vs other types of genetic amplification?
- 3) The statement was made that, "There was no significant differences between pre-cancer 3/14 (21%) with multiple 215 ecDNA species and EAC samples 23/69 (33%)...suggesting that tumors may achieve subclonal ecDNA heterogeneity early on." How do the authors... make this conclusion when only 1/25 Cambridge patients with HGD had ecDNA while the FHCRC CO cohort had 7/27?
- 4) Looking at the CO vs NCO cohorts from FHCRC, the presence of HGD is a better predictor than ecDNA for progression to EAC. Even though there is inter-pathologist variability, presence of HGD is easier to assess than ecDNA.
- 5) Figure 3 shows acquisition of additional ecDNA during progression from HGD to cancer. Were there additional patients with similar findings in sequential biopsies? Why did a patient with HGD go 5.6 years between endoscopies?
- 6) The conclusions are based on characterization of clinical specimens. Mechanistically, can ecDNA transform benign Barrett's cell lines or organoids?
- 7) Do patients with HGD and ecDNA progress to EAC faster than those that do not have ecDNA? Do cancers with ecDNA have a more aggressive histology (e.g. signet ring) or higher grade?
- 8) In clinical labs without NGS, is there another way to assess ecDNA such as FISH or ecSeg, if ecDNA is proven to be a useful biomarker?

Data and methodology, statistical analysis and mathematical models, and conclusions were appropriate. Clarification on the above would increase confidence in the role of ecDNA in BE progression to EAC. The manuscript was clear and concise and presented in an understandable manner.

Referee #3 (Remarks to the Author):

In their manuscript 'Extrachromosomal DNA in the cancerous transformation of Barrett's esophagus' Luebeck et al. use WGS sequencing data to study the role of extrachromosomal DNA (ecDNA) in the evolution of Barrett's esophagus (BE) and its progression to esophageal carcinoma (EAC). For this the authors conduct a detailed analysis of ecDNA in a cross-sectional cohort consisting of 206 cases of with different histological stages along the BE-EAC spectrum.

The authors demonstrate that the frequency of ecDNA increases with disease progression, providing a strong argument for the importance of ecDNAs in the transformation of BE. Using data from a second, longitudinal cohort consisting of two time-points from 40 progressed and 40 non-progressed BE cases, the authors show how the presence of ecDNA is associated with an increased likelihood of progression, that ecDNA gets amplified during BE transformation and that ecDNA undergoes structural changes over time. The authors also present a compelling example of the evolution of a BE towards EAC and the role of two ecDNAs using multiple timepoints in one case.

General comments:

The authors should be congratulated for their work. The paper is of high quality and certainly of interest for many readers, both in the field of cancer biology and for a wider audience. Undoubtedly,

this paper shows the importance of ecDNA in tumor evolution, an area of research that some of the authors have pioneered. This manuscript also illustrates many of the open questions of ecDNA evolution and demonstrates the importance of studying ecDNAs.

The statistical analysis appears sound, correct and appropriate throughout the paper, the quality of writing and figures is very high. However, I do have a few relevant questions and comments.

Major comments:

- The identification and reconstruction of ecDNAs is a challenging problem. The development of methods is an important avenue of research and indeed the author provide improved methods as part of this manuscript. Given this and the potential differences in purity, ploidy and coverage of the different BE and ECA samples, can the authors demonstrate that their method is sufficiently robust to such changes in sample properties and that these are not affecting the conducted analysis? What is the power to detect an ecDNA of given length and amplification level under such different conditions?
- On a similar note. Can the authors demonstrate that they are indeed able to identify ecDNA molecules correctly and that they can correctly discriminate those from fsCNAs with the WGS data used? Can the authors, for example, validate that their predictions are indeed correct using methods like FISH, PCR or long-read sequencing?
- The authors talk about the differences in the frequency of ecDNAs across disease stages and while this is relevant, it is unclear how and if this relates to other known markers. It is for example known that chromosomal instability can predict progression of BE (e.g., Paulson et al. 2009 or Killcoyne et al. 2020). Likewise, TP53 mutation status has been reported to be prognostic (e.g., Redston et al. 2021). Given their reported association with TP53 mutation status how do ecDNAs fit in this picture? What do the authors believe the causal mechanism of their observations to be? If ecDNAs are a (later) consequence of TP53 mutation status and chromosomal instability, does this mean that they are a bad prognostic marker of BE progression? What is the correlation of ecDNAs, focal CNA amplifications, chromosomal instability, and TP53 status? Are ecDNA predictive after accounting for these other variables?
- The authors comment on the average number of oncogenes per ecDNA (0.76, see line 225) and that this is more than in fsCNAs (0.52). How often are two or more oncogenes associated with an ecDNA (i.e., could you provide a supplementary figure showing this)? Are 0.76 oncogenes per amplicon more than one would expect by chance? Does this tell more about how (negative) selection might act on ecDNA molecules?
- The authors should share more data on the inferred ecDNA and fsDNA structures of different complexities to give a better idea what this summary static measures and what the actual difference in the ecDNAs complexity is caused by.
- While this study is very comprehensive, the study should cite some of the previous research on

double minutes in BE, especially where they are recognized for their clinical relevance (e.g., ERBB2, MYC, ZNF217).

Minor comments:

- While the importance of the ecDNA-2 associated genes (SOCS1, CIITA, RMI2) in P391 is very convincing, are these also highly expressed? Might the authors potentially have access to any histological for IHC staining of these genes to demonstrate the expression of these genes?

- The number of the Cambridge cohort seems to be wrong and incomplete in the method section. Should it be 51 early-stage T1, plus 88 late stage ECAs (line 302)?

- The phrase 'highly significant' on line 129 should be changed or a p-value provided (e.g., 0.0003 for one-sided NCO vs CO).

- How concordant did the authors find the purity estimates of the pathologist and those determined by sequencing (line 326)?

- What was the archived coverage, purity & ploidy distribution for WGS samples across the different disease stages and cohorts?

Author Rebuttals to Initial Comments:

Referees' comments:

Referee #1 (Remarks to the Author):

Referee 1 General Comments: “The authors have submitted an important study significantly advancing our understanding of the genomic drivers of Barrett’s esophagus and how variant amplifications may be elaborated through ecDNA mechanisms. This work is important and though the results have identified a novel biomarker for risk stratification or a novel target for intervention. The selected cohorts are well established and appropriate to address the research question. The methodological framework is sound as are the computational analyses.”

Response: We thank the reviewer for these comments.

Referee 1 point 1: *“Although the methodology is sound, I have several comments and questions on the selected cohorts that may help improve this otherwise well written manuscript:*

1) In the Abstract and Discussion, the latter which is rather brief, it would help the reader to set the frame. What is the clinical significance of ecDNA - how will this impact patient care, if true - can ecDNA be targeted directly independent of the amplified gene. Please set the clinical context and significant of this work.”

Response: We greatly appreciate the reviewer’s suggestion to provide additional clinical context and highlighting the significance of the work. In the revised MS, we have done the following:

In the revised MS, we now clearly state, that ecDNA’s are found in some of the most aggressive forms of cancer, including EAC, and that they are associated with high level oncogene expression, rapid tumor evolution, including therapeutic resistance, and significantly shorter survival in patients. We describe recent published work from our group and others, showing that ecDNAs lack centromeres, are subject to random inheritance during cell division, consequently, driving intratumoral genetic heterogeneity. We describe recent work on ecDNA dynamics, showing how the non-chromosomal inheritance promotes rapid tumor evolution leading to treatment resistance within one to two cell divisions. We further highlight that ecDNAs have highly accessible chromatin and altered cis- and trans-gene regulation, enhancing oncogenic transcriptional programs. The inclusion of this information in the introduction of the revised MS, will provide more clinical context and orient readers as to the significance of ecDNA in cancer. Further, this revised introduction now provides rationale for why readers should be interested in whether ecDNA can be found during the transition of BE to EAC, particularly if new ways of preventing or targeting ecDNA can be developed for therapy or prevention.

We have added the following sentences to the paper:

Lines 70-77: “ecDNAs are found in some of the most aggressive forms of cancer, including EAC, whose highly accessible chromatin and altered cis- and trans-gene regulation enhance oncogenic transcriptional programs^{16–20}. ecDNAs lack centromeres, and are consequently subject to random inheritance during cell division, driving intratumoral genetic heterogeneity⁶. These unique features of ecDNAs contribute to aggressive tumor growth, accelerated evolution, and therapy resistance. Patients with ecDNAs in their tumors have significantly shorter survival, even compared to other forms of genomic focal amplification³. Computational tools can detect ecDNA in sequencing (WGS) data from biopsies^{21–23}.”

Lines 310-313: “These findings shed new light on how ecDNA can arise before the development of full-blown cancer, indicating that it is not simply a late manifestation of genome instability, and raises the future possibility of earlier intervention or prevention for patients with ecDNA-containing tumors.”

Referee 1, Point 2a: “*In the Study Samples and Methodology sections please describe these cohorts in more detail. e.g. What is the minimal duration of the follow up in the non-progressor cohort who did not develop HGD/EAC in the cross sectional cohort(Cambridge)?*”

Response: We thank the reviewer for suggesting this. For the UK Non-Dysplastic Barrett’s Esophagus (NDBE) and Low-Grade Dysplasia (LGD) cohort, there was a minimum follow-up of 44 months (median 139, max 258) for the non-progressor cohort (Katz-Summercorn et al., Nat. Commun. 2022. PMID: 35301290). The minimal duration of the follow-up in the FHCC non-cancer outcome cohort was 0.88 years, and a median of 10.50 years; 8.6 years for those where the time to death was known, and 13.9 years for those with a last-known-alive status (Supplementary Table 2. We note that the FHCC study was designed to follow the natural history of patients to a validated EA endpoint; any other diagnosis was by definition not progression to EA. At the time the patients were being followed, ablation therapies were not available, nor was LGD yet recognized as having an actionable risk level. Below, please find a graphic summary of the FHCC NCO cohort, which is included as newly added Extended Data 5b and Supplementary Table 2.

Referee 1, point 2a, continued: “Why was LGD included in the non-progressor group - a diagnosis with a high degree of pathology disagreement but with a clearly defined risk of progression justifying invasive intervention (e.g ablation).”

Response: We thank the referee for pointing this out. In the revised MS, we now provide an updated Extended Data Figure 1a, show below, in which LGD and NDBE status are clearly separated. Further, we have clarified this point in the MS text and figures of the revised MS. Neither NDBE nor LGD showed evidence of ecDNA and did not progress past that phase over a median of >11 years of follow-up. Therefore, we combined the two disease states for the purposes of our statistical analysis for main Figure 2b.

Referee 1, point 2b: “In this context can you describe the review process that confirmed the pathologic diagnosis?”

Response: To address this important point, we have revised the Method section with considerably more detail. To summarize for the referee, for the Cambridge Cohort, we identified 315 suitable patients from a prospective surveillance study of >3000 pre-cancer patients for the Katz-Summercorn et al., Nat. Commun. 2022 study. These patients reflected the different grades of BE and dysplasia and for whom we had high quality frozen samples and extensive follow-up data. This included 27 patients with NDBE, 15 with LGD who never developed HGD or EAC during

follow-up, 25 patients with high-grade dysplasia (HGD), 51 patients with early-stage EAC (AJCC stage I), and 88 patients with late-stage EAC (AJCC stage II-IV) (Figure 1a) Dysplastic samples were taken from the latest follow-up endoscopy and **prior to any ablative therapy**, thus representing each patient's highest grade of disease. All biopsies underwent whole genomic (50x), transcriptomic and epigenomic (850k array) profiling according to material availability.

For the Cambridge data the International Cancer Genome Consortium (ICGC) histopathological cellularity threshold is >70% for late-stage EAC cases – this high threshold stems from the recognition that pathological assessment is crude and in practice when compared with the sequencing ground truth we know that the cellularity is in fact lower given the variable amount of stroma. This stringency is therefore necessary to ensure high data quality and minimize contamination for sensitive detection of focal amplifications from sequencing data.

Cellularity of 70% is too stringent for Barrett's cases and early-stage EAC, and we should have explained this more clearly. The Barrett's cases all had columnar epithelium containing IM in the sample. Many cases have a mix of dysplasia and IM. Strict selection criteria were implemented to ensure that only biopsies with pathology consensus agreement of the highest histological dysplasia grade were sequenced. Similarly, there are several cases in the early cancer group with intra-mucosal (T1a) carcinoma whereby cellularity is < 70% which was included.

For the pre-cancer BE cohort, any potentially suitable biopsies were reviewed independently by 2 consultant pathologists. All pathologists were blinded to the grade of the patient. Sample grade was determined by an agreement of at least two pathologists. Samples with no agreement were reviewed by the 3 pathologists together to reach a consensus. Dysplastic samples for sequencing had to have a pathological cellularity of dysplasia of >30%. For the BE adjacent to cancer, H&Es were reviewed independently by two pathologists and only dysplastic cases were reviewed by a third.

The following paragraphs have been revised in the methods section of the revised MS (beginning at line 338), as follows:

Cambridge sample selection: “We identified suitable patients from a prospective surveillance study of >3000 pre-cancer patients in the Katz-Summercorn et al., 2022 study, in which the - follow-up and methods for pathology reviewed were described. There was a minimum follow-up of 44 months (median 139, max 258) for the cohorts which did not progress to HGD/EAC, and a median follow-up of 57 months (range 0-249) for the dysplastic cohort. We analyzed WGS data from 206 patients in cross-sectional BE surveillance Cambridge cohorts with biopsy validated BE, including 27 patients with non-dysplastic Barrett's esophagus (NDBE), 15 with low-grade dysplasia (LGD) who never developed HGD or EAC during follow-up, 25 patients with high-grade dysplasia (HGD), 51 patients with early-stage EAC (AJCC stage I), and 88 patients with late-stage EAC (AJCC stage II-IV) (Figure 1a). Patients with low-grade BE and high-grade BE underwent surveillance at Cambridge University Hospitals NHS Trust and consented prospectively to a biomarker and genomic characterization study (Cell Determinants Biomarker, REC no. 01/149, BEST2 REC no. 10/H0308/71).

“Samples from patients included in the Cambridge surveillance cohorts were all treatment naïve - with no neoadjuvant chemotherapy, radiation, ablation therapies - except for two of the late-stage EAC patients 167 and 155, who had received prior therapy, as detailed in Supplementary Table 1 and Extended Data Figure 1a.

“Strict selection criteria were implemented to ensure that only the highest cellularity biopsies, with agreement of histological grade were sequenced. Potential biopsies were placed into OCT and a single section cut and stained with hematoxylin and eosin (H&E). These were reviewed by at least two consultant pathologists to assess the composition of the biopsy. All pathologists were blinded to the grade of the patient. Samples with no agreement were reviewed with a third pathologist to reach a consensus. Dysplastic samples for sequencing had to have a pathological cellularity for dysplasia of $\geq 30\%$ and labelled to be consistent with the highest pathology grade reported within the biopsy ($\geq 70\%$ percent cellularity of tumor for early-stage cancers) Non-dysplastic BE biopsies had to contain intestinal metaplasia.

“Early-stage and late-stage Cambridge EAC patients were recruited for the EAC International Cancer Genome Consortium (ICGC) study, for which samples were collected through the UK-wide Oesophageal Cancer Classification and Molecular Stratification (OCCAMS, Rec. no. 10-H0305-1) consortium. For early-stage cancer samples, cellularity $\geq 70\%$ were included, consistent with ICGC guidelines. Ethical approvals for these trials were from the East of England-Cambridge Central Research Ethics Committee. EAC samples were prospectively collected as endoscopic biopsies or resection specimens. All tissue samples were snap frozen and blood or normal squamous epithelium (at least 5cm from the tumor) were used as germline reference as previously described (Katz-Summercorn et al., 2022).”

For the FHCC cohort, we provide the following information in the methods section of the revised MS (beginning at line 447): “The histology data from FHCC are a re-analysis of the previously published cohort described in Paulson et al., 2022¹⁴. In brief, the biopsy samples which underwent WGS were not assessed for pathologic diagnosis, by design. Instead, the pathology pathological analysis was conducted from adjacent level biopsies from the esophagus, as described in Paulson et al.”

Referee 1, point 2c: *“These two (Cambridge and FHCC) cohorts are well characterized, I'm certain that this information is available, or a prior publication can be cited. For the TP-1 and TP-2 in the patient who did NOT develop cancer, what is the minimal time interval.”*

Response: We thank the referee for spurring us to include these data in the revised MS. The patients who did not progress to EA in the FHCC cohort had a mean time interval between TP-1 and TP-2 of 3.42 years, a median of 2.775 years, a minimum of 1.07 years and a maximum of 8.24 years. Please see the figure below, which for the reviewer's convenience we have re-plotted, and for which the source data is part of Supplementary Table 1 of this MS.

We have included the following line in the revised MS: (line 102):

“At least two biopsies for WGS were obtained by isolating epithelial tissue (Figure 1c) of the BE at each of the two primary study time points - timepoint 1 (TP-1) and timepoint 2 (TP-2), which were on average 2.9 years, and 3.4 years apart for CO and NCO patients, respectively (Supplementary Table 1).”

The paragraph also cites the Paulson et al., Nat. Commun. 2022 [PMID: 35484108] study as the source of this information.

Referee 1, Point 3: “Although there is a school of thought professing that all EAC originates from BE, this remains controversial. Do the authors have the granularity of clinical annotated data for the TCGA cohort (doubtful) to determine if the selected cohort were associated with BE? Or can they glean additional markers of BE from the sequencing data (e.g. TFF3) to confirm the origin of the EAC.”

Response: For the FHCC cohort, all the patients in the study had validated BE prior to progression to EA (Paulson et al., Nat. Commun. 2022). In the Cambridge cohort, we had observed that 45% of tumors were observed to have Barrett’s adjacent to the tumor location, compared to 55% without evidence the presence of BE. This estimate is based on endoscopy and pathology review as described in Sawas et al., Gastroenterology 2018 [PMID: 30165050]. A subset of us have published a recent paper suggesting that EAC does originate from BE, even in the absence of detectable Barrett’s disease following formation of EAC (Nowicki-Osuch et al., Science 2021 PMID: 34385390). Due to the invasive nature of EAC, EAC tissue may replace the progenitor BE tissue, reducing the ability to observe concurrent BE and EAC. This finding was orthogonally supported by a computational modelling study (Curtius et al., Gut 2020 PMID: 33234525).

The clinical annotation of TCGA, regrettably, is not such that the presence of BE can be ascertained. Fortunately, these TCGA patients were also involved in the Nowicki-Osuch et al., 2021 study, which further established the BE origin of the EAC tumors, using single-cell technologies.

Referee 1, Point 4: “For the Cambridge and FHCC cohorts, Do the authors have the percent of each histological diagnosis in the FFPE sections prior to sequencing in the Cambridge diagnosis? Was clinical pathological followed (i.e. the highest grade determines diagnosis even if 1% of the sample) or a more strict definition applied to have a more accurate representation of the tissue analyzed by WGS?”

Response: For the Barrett’s cases from the Cambridge cohort, strict selection criteria were implemented to ensure that only the highest cellularity biopsies, with agreement of histological grade, were sequenced. This strict definition was indeed applied to have a more accurate representation of that tissue state in WGS analysis, as asked by the reviewer. Potential biopsies were placed into OCT and a single section cut and stained with hematoxylin and eosin (H&E). These were reviewed by at least 2 consultant pathologists to assess the composition of the biopsy. All pathologists were blinded to the grade of the patient. Samples with no agreement were reviewed by the 3 pathologists together to reach a consensus. Dysplastic samples for sequencing had to have a pathological cellularity of dysplasia of >30% and labelled to be consistent with the highest pathology grade reported within the biopsy.

To provide more detail as to the percent of each histological diagnosis in the FFPE sections, as well as additional relevant information demonstrating the highest level of pathology quality control for the samples, we have visualized these data for the reviewer as a figure shown below, demonstrating the distribution ploidy, purity (or “fraction of aberrant cells” for Cambridge data) and the cellularity level of the histology. These data are now also reported in Supplementary Table 4 of the revised MS.

For the FHCC cohort, the biopsy taken for sequencing was not evaluated for histology, which would not have been possible with the way the study was designed. Instead, patient diagnosis status was defined by the highest pathologic grade found in the adjacent Barrett’s segment during a given endoscopy.

Referee 1, Point 4 (continued): “Would the authors clarify what they are trying to convey on line 306-307, do they mean that for adenocarcinoma a threshold of 70% of the epithelial cells have to be malignant to be included for sequencing? That appears rather strict. What percent of the tissue is "contaminated" by lower grade epithelia – could a supplemental table be included?”

Response:

For the Cambridge data the International Cancer Genome Consortium (ICGC) histopathological cellularity threshold is >70% for late-stage EAC cases – this high threshold stems from the recognition that pathological assessment is crude and in practice when compared with the sequencing ground truth we know that the cellularity is in fact lower given the variable amount of stroma. This stringency is therefore necessary to ensure high data quality and minimize contamination for sensitive detection of focal amplifications from sequencing data.

Cellularity of 70% is too stringent for Barrett's cases and early-stage EAC, and we should have explained this more clearly. The Barrett's cases all had columnar epithelium containing IM in the sample. Many cases have a mix of dysplasia and IM. Strict selection criteria were implemented

to ensure that only biopsies with pathology consensus agreement of the highest histological dysplasia grade were sequenced. Similarly, there are several cases in the early cancer group with intra-mucosal (T1a) carcinoma whereby cellularity is < 70% which was included. We have amended the text as follows:

Line 357: “Strict selection criteria were implemented to ensure that only the highest cellularity biopsies, with agreement of histological grade were sequenced. Potential biopsies were placed into OCT and a single section cut and stained with hematoxylin and eosin (H&E). These were reviewed by at least two consultant pathologists to assess the composition of the biopsy. All pathologists were blinded to the grade of the patient. Samples with no agreement were reviewed with a third pathologist to reach a consensus. Dysplastic samples for sequencing had to have a pathological cellularity for dysplasia of $\geq 30\%$ and labelled to be consistent with the highest pathology grade reported within the biopsy ($\geq 70\%$ percent cellularity of tumor for early-stage cancers) Non-dysplastic BE biopsies had to contain intestinal metaplasia.”

Regarding the percent of contaminated tissue, please see the response and figure above, which are now presented in Supplemental Table 4 of the revised MS.

Referee 1, point 5: “As alluded to above, the discussion is rather brief and does not put important results in context with the larger clinical picture and unmet clinical needs for Barrett’s esophagus. Please expand the discussion and include a brief clinical implication of the work. How can early detection of ecDNA in non-dysplastic BE influence the management of patients? Risk stratification for shorter/longer surveillance intervals? early invasive intervention (e.g. ablation), targeting ecDNA directly to prevent/revert/reprogram “damned” mucosa?”

Response: We thank the reviewer for providing the opportunity to expand on significance in the discussion section. We have revised the text of the final paragraph to highlight that the discovery of ecDNA during HGD, its strong association with cancer development, its selection and evolution over time, as well as the finding of its link to TP53 alteration and to the presence of immunomodulatory genes on ecDNAs, provides important new biological insight. It also suggests that rather than focusing on ecDNA as a “biomarker” of cancer development, it is rather an opportunity for earlier intervention, either as therapy or prevention, as these strategies are developed.

The following sentences have been added to the discussion of the revised MS (line 310):
“These findings shed new light on how ecDNA can arise before the development of full-blown cancer, indicating that it is not simply a late manifestation of genome instability, and raises the future possibility of earlier intervention or prevention for patients with ecDNA-containing tumors.”

We thank the Referee 1 for these highly constructive and insightful comments. Addressing them has strengthened the manuscript.

Referee #2 (Remarks to the Author):

This manuscript by Luebeck et al is the first to examine ecDNA frequency and contribution to pathogenesis in the Barrett metaplasia, dysplasia, adenocarcinoma progression sequence. Using two separate large cohorts, the authors showed that ecDNA frequency increased from metaplasia to early and late stage cancer using WGS and Amplicon analysis. In longitudinal samples from a single patient, acquisition of a second ecDNA was associated with progression to adenocarcinoma from HGD associated with an ecDNA. Finally analysis of ecDNA in cancer compared to pre-cancer lesions demonstrated that cancer ecDNA were more complex and coded for oncogenes not seen on non-ecDNA amplifications and immunomodulatory genes. In one cohort that followed patients over time, the presence of ecDNA predicted progression to EAC.

Major comments:

Referee 2, Point 1a: “Since ecDNA generation could occur due to therapeutic interventions (e.g. ablation, chemotherapy, radiation) and resulting DNA damage, especially in more advanced disease, further description of the patient cohorts would be useful. The Cambridge cohort of 42 non-dysplastic BE and 25 HGD was described as a surveillance cohort in the text, but in Figure 1 the HGD patients were described as having received therapy. No description of treatment was provided for the FHCRC cohort, were these patients with HGD followed without intervention?”

Response: We thank the referee for raising this critical question. Please note, only two patients in the Cambridge cohort received treatment, and both had late-stage EAC and had received prior therapy, which is noted in Extended Data Figure 1a. The rest of the patients, nearly all of them, were treatment-naïve. We have clarified this point in the figure and text. Further, all of the patients in the FHCC cohorts were treatment-naïve. We have updated this information in the Methods section of the revised MS, to address this critical point, as follows:

Cambridge cohort (line 352): “Samples from patients included in the Cambridge surveillance cohorts were all treatment naïve - with no neoadjuvant chemotherapy, radiation, ablation therapies - with the exception of two of the late-stage EAC patients 167 and 155, who had received prior therapy, as detailed in Supplementary Table 1 and Extended Data Figure 1a.”

FHCC cohort (line 424): “All of the samples collected for the FHCC study were from treatment-naïve patients.”

Referee 2, Point 1b: “What was the length of follow-up of the Cambridge NDBE cohort and the FHCRC NCO cohorts (9.6 years was provided for only a subset)?”

Response: For the Cambridge cohort, there was a minimum follow-up of 44 months (median 139, max 258) for the non-progressors and a median follow-up of 57 months (range 0-249) for the dysplastic

cohort. These details were previously published in Katz-Summercorn et al., 2022 [PMID: 35301290], and we have cited the follow-up numbers on line 340) of the revised MS:

“There was a minimum follow-up of 44 months (median 139, max 258) for the cohorts which did not progress to HGD or EAC, and a median follow-up of 57 months (range 0-249) for the dysplastic cohort.”

For the FHCRC NCO cohort, the length of follow up for the NBDE patients is shown below, in new Supplementary Table 2 and Extended Data Figure 5b (shown below) of the revised MS. These additional information are also provided in the supplementary methods section. As a brief thumbnail for the referee, we note:

1. 13.86 years - median to last known alive
2. 8.58 years - median years to death from other causes
3. 10.50 years - median length of follow-up (combined last known alive and years to death)
4. 0.88 years - minimum follow-up duration for NCO
5. 85.0% - percentage of FHCC NCO patients followed >5 years

Also, please note that this is different than the time between TP-1 and TP-2 for the NCO patients, as shown in response to Referee 1, point 2c, above.

Referee 2, Point 2: “How accurate are AmpliconArchitect and AmpliconClassifier at identifying ecDNA vs other types of genetic amplification?”

Response: The accuracy of the original versions of AmpliconArchitect (AA) and AmpliconClassifier (AC) at identifying ecDNA, relative to other types of genetic amplifications, was published in Kim et al., Nature Genetics, 2020. In that initial publication, sensitivity of ecDNA detection on a panel of 67 genes from 42 cell-lines with cytogenetically validated ecDNA was 83%. Of note, the tools have advanced since then and have even better sensitivity of detection. Therefore, to address the Referee’s question, we evaluated AA/AC predictions using cell-lines for which whole genome sequencing and metaphase FISH were both available (>1,600 total images).

Metaphase FISH is considered a gold-standard because when the chromosomes align during cell division, it is possible to accurately detect whether amplified oncogenes are on ecDNA or on chromosomes. This can't be done with tissue samples or publicly available WGS data or exome data. Therefore, we performed this analysis, as follows. As shown, in the figure below, AA and AC are quite good at detecting ecDNAs relative to other types of amplifications – precision of 76.5% and a sensitivity (recall) of 87%. We observed the “false positives” largely arise from ecDNAs that have reintegrated into chromosomal HSRs at lower copy number, but for which the “fossil record” of an ecDNA can still be detected. Please note, that during in vitro cell culture, there is a tendency for ecDNA reintegrate into HSRs, so our estimate of precision is extremely conservative.

We further applied an orthogonal approach to directly address the reviewer’s question by examining Oxford Nanopore technology (ONT) data from an EAC tumor sample (not present in this study) which contained two focal amplifications. These data are part of an upcoming manuscript by the Cambridge team, and thus are not part of this manuscript. AA revealed an ecDNA focal amplification (A03.0_amplicon1) and a BFB focal amplification (A03.6_amplicon1), shown below. **REDACTED**. Due to the complex structures of both focal amplification classes, distinguishing them is computationally challenging.

REDACTED

We tested if the classification of AA for the ecDNA and BFB could be validated by the circularity of the resulting de novo assemblies of the ONT data. We found that the **REDACTED** ecDNA amplicon formed a cyclic assembly structure compared to a **REDACTED** BFB which assembled segment. When comparing the AA result to the Oxford Nanopore assembly, we found that in the ecDNA case, the internal duplication of **REDACTED** was captured in the circular contig, where two spanned a region with 2x as much ONT coverage **REDACTED**. For the BFB structure on the other hand, while the ONT assembly was non-cyclic, it failed to capture the foldback SVs characteristic of BFB structure. Thus, while the Nanopore assemblies support our predictions, they are not always capable of resolving multiple copies of genomic segments inside focal amplifications (particularly BFB) and may struggle to handle long internal duplications seen in focal amplifications, without additional method development. We are actively working on ONT-based detection, and hope to publish those results in the near future.

REDACTED

Referee 2, Point 3: "The statement was made that, "There was no significant differences between pre-cancer 3/14 (21%) with multiple 215 ecDNA species and EAC samples 23/69

(33%)...suggesting that tumors may achieve subclonal ecDNA heterogeneity early on." How do the authors make this conclusion when only 1/25 Cambridge patients with HGD had ecDNA while the FHCRC CO cohort had 7/27?"

Response: We appreciate the referee's concern that this statement may be confusing to readers. Therefore, in the revised MS, we have modified the text as follows (line 241):

"26/83 (31%) of ecDNA-positive samples from the combined study sources contained more than one species of ecDNA (Figure 4e), enabling multiple oncogene amplifications. Multiple ecDNA species could also be detected in BE HGD samples from patients that progressed to EAC (Supplementary Table 3), raising the possibility that tumors may achieve subclonal ecDNA heterogeneity early on, and that competition between multiple distinct ecDNAs could potentially play a role in EAC evolution."

Referee 2, Point 4: "Looking at the CO vs NCO cohorts from FHCRC, the presence of HGD is a better predictor than ecDNA for progression to EAC. Even though there is inter-pathologist variability, presence of HGD is easier to assess than ecDNA."

Response: We agree with the referee. The purpose of this study is not to develop a better biomarker for determining which patients with BE will go on to cancer. Members of our team, and others, have already shown that HGD, as well as copy number alterations and TP53 alteration, are strongly associated with EAC development. Rather, our study sheds unique new light and biological insight into how ecDNA can be a molecular event that arises before cancer development. We show that it can occur far earlier than anticipated. These results clearly show that ecDNA is not simply a late manifestation of genome instability. Further, the strength of the association between ecDNA and cancer development, its clear selection during cancer development and evolution, and the surprising contents including immunomodulatory genes, shed new insight into the role of ecDNA in cancer and suggest new opportunities for therapeutic intervention and potentially prevention in the future. We have made these points much clearer in the discussion section of the revised MS. We thank you the referee for providing this opportunity to highlight and focus on the key points of our paper.

We have modified the discussion, line 310, as follows: "These findings shed new light on how ecDNA can arise before the development of full-blown cancer, indicating that it is not simply a late manifestation of genome instability, and raises the future possibility of earlier intervention or prevention for patients with ecDNA-containing tumors."

Referee 2, Point 5: "Figure 3 shows acquisition of additional ecDNA during progression from HGD to cancer. Were there additional patients with similar findings in sequential biopsies? Why did a patient with HGD go 5.6 years between endoscopies?"

Response: The design of the FHCC study was to evaluate patients at two time points, as close to their initial baseline endoscopy as possible and, in the case of progressors, at the timepoint EAC was diagnosed. Therefore, WGS was only available from samples at these two time points. Patient 391, whose samples are featured in Figure 3, underwent 18 endoscopies with histology biopsies during the study period (Reported in Supplementary Data File 2 of Paulson et al. Nat. Commun., 2022). Patient 391 was unique in that samples from additional time points, including the resected tumor, also underwent WGS, allowing us to follow the evolution of ecDNA over time and space in a patient that progressed to EAC. We also note that the FHCC study began years ago, when patients with HGD were frequently watched as part of standard treatment guidelines. This feature of the study's design, in fact, created the unique opportunity to observe the "natural history" of EAC development from BE in this particular patient who developed ecDNA, as shown in Figure 3. This case was, fortuitously, highly informative, additional "natural history" samples with WGS data were not available for analysis from other patients.

Referee 2, Point 6: "The conclusions are based on characterization of clinical specimens. Mechanistically, can ecDNA transform benign Barrett's cell lines or organoids?"

Response: We thank the referee for this great question. We are thinking very much along the same lines, and this is precisely the kind of long-term experiment, with investment in modeling that we are planning as part of our Cancer Grand Challenges Program. However, this set of experiments would require a very significant commitment to constructing such an in vivo model. The idea of using cell lines or organoids is also highly appealing. Currently, it's not technically possible to synthesize/purify intact ecDNA for transfection. We anticipate that this will be doable at some point in the future. Our correlative data, as well as work from our recently published paper in Nature Genetics (Lange et al., 2022, PMID: 36123406), suggest a direct causal role in driving ecDNA-containing tumor growth. However, testing its role in initial tumor formation will be dependent on developing these technologies for putting whole ecDNAs into non-cancer cells or organoids in the right genetic contexts, hopefully in the near future. Consequently, we are careful to state clearly in our paper, that our results show that ecDNA can arise during HGD, and that it is selected for during cancer evolution, and that testing for mechanistic proof of causality in cancer development from HGD will be an exciting set of experiments for the future.

Referee 2, Point 7: "Do patients with HGD and ecDNA progress to EAC faster than those that do not have ecDNA?"

Response: This is an important question. However, the design of the FHCC Paulson et al. study does not allow us to analyze timing differences between CO and NCO, etc. because the distribution of times from TP-1 to TP-2 were selected to be as equal as possible (within what was possible clinically). The mean time for the NCO patients was 3.4 years and the mean time for CO patients was 2.9 years, and there was no significant difference between the two groups (Mann Whitney U-test, p-value=0.057).

While this timing data is published previously in Paulson et al., we have re-visualized it for the reviewer's convenience below:

Similarly, the Cambridge cohort study is not designed to answer that question, as patients with HGD are subsequently treated after biopsies are taken and thus natural disease progression cannot be observed.

Referee 2, Point 7 continued: “Do cancers with ecDNA have a more aggressive histology (e.g. signet ring) or higher grade?”

Response: None of the Cambridge EAC cases had signet ring histology reported and no obvious differences in tumor histology were evident. However, to address the referee's question more deeply, we examined the cellularity percentage of EAC in the tumor sample histology data from the Cambridge cohort (reproduced below on left, and now shown in Supplementary Figure 4c). ecDNA+ tumors showed a slightly higher tumor cellularity in the sample 81% vs 76%. However, the difference was not statistically significant as evaluated by a Mann-Whitney U test ($p=0.057$). Given the borderline value, it is possible that ecDNA+ cancers are more cellular. However, we are not powered in this study to answer that question and a larger-scale study with greater statistical power would be more revealing.

More directly, we did find tumors with ecDNA in the Cambridge cohort were more strongly associated with a higher cancer stage (stage II+ vs. stage I) than samples without ecDNA (Fisher's exact test, $p=0.027$, one-sided test). We have added a figure illustrating this finding to the revised manuscript (Extended Data Figure 1b, reproduced below).

Referee 2, Point 8 “In clinical labs without NGS, is there another way to assess ecDNA such as FISH or ecSeg, if ecDNA is proven to be a useful biomarker?”

Response: This is a great question of high practical value. We are currently developing new tools to cytogenetically analyze samples for ecDNA presence, both in mitotic cells at metaphase (Rajkumar et al., iScience 2019 PMID: 31706138), but also in interphase cells (in progress work). This will be, we hope, achievable in routinely processed, FFPE samples. The technology is called ecSegi (note, “i” stands for interphase). We cannot include it in the revised MS, but for the reviewer’s interest, please see a screenshot below, of a readout from ecSegi.

REDACTED

ecDNA+ cells (COLO320DM, middle) show multiple probes per cell and a high heterogeneity while HSR cells (COLO320HSR, left) or ecDNA- are typically a constant signal per cell with small number of distinct spots. A metaphase nucleus is also observed, for explanation purposes, but the interpretation is made solely using interphase cells.

On tissue cultures of cell lines, where the ecDNA status is known, the ecSegi method predicted ecDNA+ and ecDNA- lines **REDACTED**. The first 2 panels show the differential signal between ecDNA+ cells from COLO320DM and ecDNA- cells from COLO320HSR-. The right panel shows the distribution of automatically detected FISH spots in a multitude of cell-lines, confirming that ecDNA+ have higher number of FISH signals and greater heterogeneity from cell to cell.

While imaging data is not available for this Barrett’s data, the question asked by the reviewer is very important. We believe publication of our current manuscript will establish greater importance for the ecSegi method when ready for publication.

Referee 2, Point 9 “Data and methodology, statistical analysis and mathematical models, and conclusions were appropriate. Clarification on the above would increase confidence in the role of ecDNA in BE progression to EAC. The manuscript was clear and concise and presented in an understandable manner.”

Response: We thank the Referee 2 for these highly constructive and insightful comments. Addressing them has strengthened the manuscript.

Referee #3 (Remarks to the Author):

In their manuscript ‘Extrachromosomal DNA in the cancerous transformation of Barrett’s esophagus’ Luebeck et al. use WGS sequencing data to study the role of extrachromosomal DNA (ecDNA) in the evolution of Barrett’s esophagus (BE) and its progression to esophageal carcinoma (EAC). For this the authors conduct a detailed analysis of ecDNA in a cross-sectional cohort consisting of 206 cases of with different histological stages along the BE-EAC spectrum.

The authors demonstrate that the frequency of ecDNA increases with disease progression, providing a strong argument for the importance of ecDNAs in the transformation of BE. Using data from a second, longitudinal cohort consisting of two time-points from 40 progressed and 40 non-progressed BE cases, the authors show how the presence of ecDNA is associated with an increased likelihood of progression, that ecDNA gets amplified during BE transformation and that ecDNA undergoes structural changes over time. The authors also present a compelling example of the evolution of a BE towards EAC and the role of two ecDNAs using multiple timepoints in one case.

Referee 3, General comments:

“The authors should be congratulated for their work. The paper is of high quality and certainly of interest for many readers, both in the field of cancer biology and for a wider audience. Undoubtedly, this paper shows the importance of ecDNA in tumor evolution, an area of research that some of the authors have pioneered. This manuscript also illustrates many of the open questions of ecDNA evolution and demonstrates the importance of studying ecDNAs. The statistical analysis appears sound, correct and appropriate throughout the paper, the quality of writing and figures is very high. However, I do have a few relevant questions and comments.”

Response: We are most grateful to the referee for these comments.

Referee 3, Point 1 “The identification and reconstruction of ecDNAs is a challenging problem. The development of methods is an important avenue of research and indeed the author provide improved methods as part of this manuscript. Given this and the potential differences in purity, ploidy and coverage of the different BE and ECA samples, can the authors demonstrate that their method is sufficiently robust to such changes in sample properties and that these are not affecting the conducted analysis?”

Response: This is a great question. Indeed, the high copy number of ecDNA in many samples helps us, as it raises the signal above the noise. In fact, in the original studies which introduced

AmpliconArchitect (AA) (Turner et al., Nature 2017, Deshpande et al., Nat. Commun. 2019), we had chosen ultra-light WGS coverage lower than 1x, and still identified ecDNA reliably. We address each of the referee's points below.

We first examined sequence coverage of BAM files. In the FHCC data, no significant difference between the coverage in ecDNA+/- cases was detected, as shown below and in Supplementary Figure 4d of the revised MS.

Of note, in discovery of ecDNA using AA on ultra-low WGS data in the Turner et al., Nature 2017 and Deshpande et al., Nat. Commun. 2019, ecDNA was robustly discovered using WGS data with under 1x coverage and validated by a large-scale FISH and DAPI-based imaging analysis. The locally high copy number of the ecDNA enable analysis of focal amplifications even when global sequencing coverage on normal regions is low. Therefore, it is not surprising that we again find ecDNA detection and coverage are unrelated.

We also analyzed the estimated purity and ploidy values for the FHCC CO and Cambridge EAC samples to search for differences in the detection of ecDNA with these variables, as shown in the figure below, which is now included as Supplementary Figure 4 of the revised MS.

Indeed, we found that in both the Cambridge cohort EAC and FHCC CO groups, ecDNA was associated with higher ploidy. However, the difference in ploidy for ecDNA+ and ecDNA- samples is expected as both increased ploidy and ecDNA co-occur with genome instability as a confounding factor.

We next evaluated the differences in cellularity (UK EAC histology) and purity (estimated from FHCC CO sequencing), as well as aberrant cell fraction (estimated from UK EAC sequencing). In all three comparisons, we did not find a significant statistical association between these factors and the presence of ecDNA, suggesting that our ecDNA detection is not biased by those variables.

These data are also summarized in the table below and included in the revised MS as Supplementary Table 4. “MWU” indicates Mann-Whitney U test.

Feature	Source	Mean values	P-value (MWU)
Coverage (FHCC)	Sequencing	74.2 ecDNA-, 74.0 ecDNA+	0.96
Ploidy (UK EAC)	Sequencing	2.61 ecDNA-, 3.19 ecDNA+	0.00023 ***
Ploidy (FHCC CO)	Sequencing	2.26 ecDNA-, 2.42 ecDNA+	0.021 *
Cellularity (UK EAC)	Histology	75.7 ecDNA-, 81.2 ecDNA+	0.057
Purity (FHCC CO)	Sequencing	0.83 ecDNA-, 0.85 ecDNA+	0.12
Aberrant cell fraction (UK EAC)	Sequencing	0.51 ecDNA-, 0.49 ecDNA+	0.93

We have added the following to the Supplementary Information section of the revised MS (Supplementary Info, page 9 “Association of ecDNA status to purity, cellularity and ploidy”):

“We analyzed the relationship of ecDNA to the purity of the samples and also to genomic instability-related events (Supplementary Table 4). We found no significant relationships between ecDNA status and estimated purity for the Cambridge EAC or FHCC CO study samples (Mann-Whitney U test, p-values = 0.93, 0.12, respectively, Supplementary Figure 4a-b). With histology-based cellularity estimates available for the Cambridge study samples, neither did we find a relationship between tumor cellularity and WGS-based ecDNA status (Mann-Whitney U test p-value=0.057, Supplementary Figure 4c). Lastly, we found no significant relationship between ecDNA status with WGS sequence coverage (Mann-Whitney U test, p-value=0.96, Supplementary Figure 4d), suggesting that sample-specific properties did not significantly influence the discovery of ecDNA with these samples.

“However, when we examined estimated ploidy in relation to ecDNA status, we found significantly higher ploidy in ecDNA+ samples (Mann-Whitney U test, p-values= 2.3×10^{-4} and 0.021 for

Cambridge and FHCC respectively, Supplementary Figure 4e-f), suggesting a link between ecDNA and genomic instability.”

Referee 3, Point 1 continued “What is the power to detect an ecDNA of given length and amplification level under such different conditions?”.

Response: To answer this question, we launched a purity simulation study to benchmark AmpliconSuite’s performance.

We selected 10 cancer cell lines – 6 EAC cell lines from Contino et al., F1000Res 2017, PMID: 28716721, and 4 from CCLE, for which we had previously predicted ecDNA. Across those cell lines AmpliconClassifier predicted 43 distinct ecDNAs, ranging in copy number from 4.8 to 271.3. Since low-copy number ecDNA are presumably the most affected by diminished purity, 37/43 (86%) of the ecDNA in the simulation study contained copy numbers under 20. By comparison, in our study, 87.5% of ecDNAs we found also had ecDNA CN under 20. The latter, however, reflects the copy number unadjusted for diminished purity, and thus the true copy numbers are likely higher.

We mixed hg38-aligned cancer cell line whole-genome sequencing reads with reads from hg38-aligned diploid NA12878 cells to form 29 different purity levels between 4.8% purity and 100% purity. All mixed bam files had coverage >10 and were then subsequently run using AmpliconSuite-pipeline with the same parameters used in this manuscript. In brief, AmpliconSuite-pipeline was used to detect focal amplification seeds using CNVKit (default parameters for AmpliconSuite and CNVKit). The seeds were then passed to AmpliconArchitect alongside the bam files and the default downsampling to coverage 10x was used by AmpliconArchitect so that all simulated purity levels and for all ecDNA had the same coverage during analysis.

We then used AmpliconClassifier (AC) to detect ecDNA from the resulting outputs using the default threshold of copy number > 4.5. A sample was assumed to have correctly identified ecDNA if it produced an ecDNA classification overlapping the original cell line’s ecDNA locations in the genome. We plotted the relationship of the original copy number, purity and ecDNA detection status below.

We found 71.8% of theoretically detectable ecDNA were recovered across all purity levels. We plotted the sensitivity of the method against the simulated purity and observed the sensitivity of ecDNA recovery was higher for higher levels of purity. Importantly, in the FHCC CO samples with ecDNA, 91% of samples had purity higher than 75%, for which the sensitivity of ecDNA detection on theoretically detectable ecDNA was ~80% and higher. Perfect recovery of ecDNA is impossible with short read sequencing, and more challenging with lower purity.

In the FHCC NCO samples, where ecDNA was rarely found, we found the purity was significantly higher than the purity of FHCC CO samples, which had abundant ecDNA, suggesting that we are not systematically under-calling ecDNA in the non-cancer samples based on purity.

Regarding the power to detect ecDNA of a given length – the AmpliconSuite-pipeline utilizes a length cutoff of 10kbp and higher for detection of focal amplifications.

Referee 3, Point 2: “On a similar note. Can the authors demonstrate that they are indeed able to identify ecDNA molecules correctly and that they can correctly discriminate those from fscNAs with the WGS data used? Can the authors, for example, validate that their predictions are indeed correct using methods like FISH, PCR or long-read sequencing?”

Response: We thank the referee for raising this important point, which was similar to the one raised by referee 2, point 2. Please see our response to that referee’s comment, shown above, as well. To summarize, the accuracy of these tools had been previously described in Kim et al (Nature Genetics). We took a panel of 67 genes from 42 cancer cell-lines where the number of ecDNA per cell had been quantified cytogenetically (Turner et al., Nature 2017), and whole genome sequencing data was available. We used this in order to benchmark the sensitivity and specificity of our methods. See response to referee 2, point #2. The tool has also continued to evolve and improve since the 2020 paper. Therefore, to address this important question, we compared AA/AC predictions using cell-lines for which whole genome sequencing and metaphase FISH were both available. In the figure below, we show the performance of AA/AC. The results suggest a precision of 76.5% and a recall of 87% in identifying ecDNA. Please note, that the precision is over penalized because in cell-lines growing in culture, many ecDNA reintegrate to form homogeneously staining regions (HSRs), retaining the “fossil record” of having arisen as ecDNA.

To provide orthogonal confirmation and further address the referee's question, we examined Oxford Nanopore technology (ONT) data from an EAC tumor sample which contained two focal amplifications. These data are part of an upcoming manuscript by the Cambridge team, and thus are not part of this manuscript. Summarizing from our response to Referee 2, point #2, AA revealed an ecDNA focal amplification (A03.0_amplicon1) and a BFB focal amplification (A03.6_amplicon1), shown below. The ecDNA focal amplification showed internal duplication of a genome segment containing REDACTED while the BFB showed amplification of a region containing REDACTED.

REDACTED

We tested if the classification of AA for the ecDNA and BFB could be validated by the circularity of the resulting de novo assemblies of the ONT data. We found that the REDACTED ecDNA amplicon formed a cyclic assembly structure compared to a REDACTED BFB which assembled into a linear segment. When comparing the AA result to the Oxford Nanopore assembly, we found that in the ecDNA case, the internal duplication of MET was captured in the circular contig, where two loops spanned a region with 2x as much ONT coverage REDACTED. For the BFB structure on the other hand, while the ONT assembly was non-cyclic, it failed to capture the foldback SVs characteristic of BFB structure. Thus, while the Nanopore assemblies support our predictions, they are not capable of resolving multiple copies of genomic segments inside focal amplifications (particularly BFB) and also handle long internal duplications seen in focal amplifications, without

additional method development. We are actively working on ONT based detection, and hope to publish those results in the near future.

REDACTED

Regarding the question of FISH validation – we are currently working on a suite of new tools for determining ecDNA status from FISH analysis of FFPE samples. Please see response to Referee 2, point 8. This approach has been developed to use a machine-learning approach to identify ecDNA location on interphase cells, hence, it is called “ecSegi”. We are not yet ready to include it in this MS or publish it, but it is work in progress. We thank the referee for raising these questions.

Referee 3, Point 3a: “The authors talk about the differences in the frequency of ecDNAs across disease stages and while this is relevant, it is unclear how and if this relates to other known markers. It is for example known that chromosomal instability can predict progression of BE (e.g., Paulson et al. 2009 or Killcoyne et al. 2020). Likewise, TP53 mutation status has been reported to be prognostic (e.g., Redston et al. 2021). Given their reported association with TP53 mutation status how do ecDNAs fit in this picture?”

Response: To address this important question and augment the existing data shown in Supplementary Fig. 6 of the initial submission (Now Extended Data Figure 6), showing a strong association between TP53 alteration and ecDNA, we conducted additional analyses to examine the relationship between TP53 alteration and either chromothripsis or whole genome doubling. As shown in the figure at the top of the following page, which is included as Extended Data Figure 6c-f of the revised MS, we show, as anticipated, a relationship between TP53 alteration and both features, consistent with the role for TP53 alteration in genome instability.

Previous work, which we have cited in the revised manuscript (Campbell et al., Nature 2020, Shoshani et al., Nature 2020, Rosswog et al. Nat. Genet. 2021, Ly et al., Nat. Genet. 2019, Stephens et al., Cell 2011, Umbreit et al. Science, 2020, Nones et al. Nat. Commun. 2014) demonstrates that chromothripsis can give rise to ecDNA. However, we have also shown that there appear to be other mechanisms towards ecDNA formation, which may also rely on TP53 alteration, consistent with the relatively weaker association between chromothripsis and ecDNA. Therefore, future studies will be needed to further elucidate the mechanisms that arise that contribute to ecDNA formation and maintenance in the presence of TP53 alteration.

We have added the following to the revised MS (line 178):

“Whole-genome duplication (WGD) and chromothripsis have ties to genome instability, and those mechanisms may contribute to ecDNA formation. In the FHCC samples we found WGD and chromothripsis were significantly associated with TP53 alteration (Extended Data Figure 6c-d), indicative of its role in mediating genomic stability. However, many of the samples with ecDNA did not show evidence of chromothripsis or whole genome duplication (Extended Data Figure 6e-f), indicating additional mechanisms of ecDNA formation following TP53 alteration.”

Referee 3, Point 3b: “What do the authors believe the causal mechanism of their observations to be? If ecDNAs are a (later) consequence of TP53 mutation status and chromosomal instability, does this mean that they are a bad prognostic marker of BE progression? What is the correlation of ecDNAs, focal CNA amplifications, chromosomal instability, and TP53 status? Are ecDNA predictive after accounting for these other variables?”

Response: Clearly, TP53 alteration, chromosomal instability and amplifications are not independent and isolated events. Indeed, large-scale genomic instability can almost be viewed as a proxy for the effects of TP53 alteration, including the effects of various mechanisms of gains and losses. We believe that TP53 alteration is essential for tolerance to chromosomal instability

and provides an increase chance of gaining additional copies of oncogene containing genomic sequences, including in EAC. Indeed, in the response to point 3a above, we found that after controlling for TP53 status, there was not a significant enrichment of ecDNA with chromothripsis status. This may be particularly true for the formation of ecDNA based on the episomal model (Storlazzi et al., Hum Mol Genet. 2006, PMID: 16452126), in addition to ecDNA formation through chromothripsis (Shoshani, et al. Nature 2020, PMID: 33361815). We note the recent study from Scott Lowe's group (Baslan et al., Nature, 2022, PMID: 35978189) demonstrating an orderly and deterministic sequence of genomic events (including copy number amplifications) following TP53 loss, giving rise to pancreatic ductal adenocarcinoma in a mouse genetic model. Our findings based on these clinical data raise the possibility that TP53 may be a critical initiating event that can start the process toward ecDNA formation in the development of EAC from BE. Future mechanistic studies will be needed to better understand this process.

Referee 3, Point 4: “The authors comment on the average number of oncogenes per ecDNA (0.76, see line 225) and that this is more than in fsCNAs (0.52). How often are two or more oncogenes associated with an ecDNA (i.e., could you provide a supplementary figure showing this)? Are 0.76 oncogenes per amplicon more than one would expect by chance? Does this tell more about how (negative) selection might act on ecDNA molecules?”

Response: We recently showed (Lange et al., Nat. Genet. 2022, PMID: 36123406), that ecDNAs, including those containing oncogenes, are under strong selection pressure. We used CRISPR-C (Møller et al., Nucleic Acids Res. 2018, PMID: 30551175) and generated DHFR-containing ecDNA in Hap1 cells. Without methotrexate selection, circular DHFR disappeared relatively rapidly before reaching neutral selection. The chromosomal scar remained unchanged, consistent with neutral selection model. In contrast, in the presence of methotrexate, DHFR copy number rose dramatically in a dose-dependent fashion, consistent with positive selection. We further combined evolutionary theory, simulations, and modeling of observations from cell lines and clinical samples, confirming that ecDNAs are under strong selection, and further, used CRISPR to genetically delete the oncogenes on ecDNA, resulting in precipitous cell death (Lange et al., Nat. Genet. 2022). Taken together, these data all indicate that ecDNAs, and the genes encoded on them, are under strong selection pressure. Consequently, it is not surprising that ecDNAs are enriched for oncogenes, and that 33% of ecDNAs contain more than one oncogene. Please see the new figure panels below from Extended Data Figure 9.

Given our finding that ecDNAs increase in copy number during progression from HGD in BE to EAC, the preferential appearance of (multiple) oncogenes is consistent with strong positive selection driving the copy number gains.

Regarding the ecDNAs that lack known oncogenes – our additional recent work (Hung et al., Nat. Genet. 2022, PMID: 36253572) suggests the presence of ecDNAs that lack oncogenes, but which are enriched for enhancers that interact with the promoters of oncogenes housed on other ecDNAs in hubs (Hung et al., Nature 2021, PMID: 34819668). This finding reveals a new mechanism of oncogenic transcription generated by ecDNA interactions in trans, highlighting the combinatorial power of ecDNA interactions. Further, our finding, of immunomodulatory genes encoded on ecDNA, as shown in Extended Data Figure 10 of the revised MS, reveals the potential gain of function activities of non-oncogenes, including regulatory sequences, that can be housed on ecDNA to enhance fitness, and which appear to be under positive selection.

Referee 3, Point 5:” The authors should share more data on the inferred ecDNA and fsDNA structures of different complexities to give a better idea what this summary static measures and what the actual difference in the ecDNAs complexity is caused by.”

Response: We are grateful for this terrific suggestion. We have created a supplementary figure that provides examples of the complexity scores so that readers can have a better understanding of how these complexity scores work. The difference in complexity can happen due to multiple reasons. However, one important reason is that as ecDNA replicate, their structure might change through deletion or recombination with other genomic segments, and also reintegration of ecDNA into the chromosome. This results in increases complexity of structural variation, which is quantified by our complexity score, as shown below in Supplementary Figure 3 of the revised MS.

Number of CN segments in ecDNA region: 1
AA genome paths (segments and length-scaled weight):
 Two paths overlapping ecDNA region:
 - Path 1: cyclic, weight=0.52
 - Path 2: non-cyclic, weight=0.48

Low complexity case: score 0.69

Complexity from CN segments = $\log(1) = 0$

Complexity of non-ecDNA-like paths
 = $-0.48 \times \log(0.48) = 0.34$

Complexity of ecDNA-like paths
 = $-0.52 \times \log(0.52) = 0.35$

Total complexity score:
 = complexity from number of segments +
 ecDNA complexity + residual complexity +
 = $0 + 0.34 + 0.35$
 = **0.69**

Number of CN segments in ecDNA region: 10
AA genome paths (segments and length-scaled weight):
 Eight paths overlapping ecDNA regions:
 - Two cyclic paths, both in top 80% of weights: 0.21, 0.07
 - Six non-cyclic paths, combined weight = 0.72

Medium complexity case: score 3.05

Complexity from CN segments = $\log(10) = 2.30$

Complexity of non-ecDNA-like paths
 = $-0.72 \times \log(0.72) = 0.24$

Complexity of ecDNA-like paths
 = $-0.21 \times \log(0.21) + -0.07 \times \log(0.07) = 0.51$

Total complexity score:
 = $2.30 + 0.24 + 0.51$
 = **3.05**

Number of CN segments in ecDNA region: 84
AA genome paths (segments and length-scaled weight):
 Twenty-four paths overlapping ecDNA regions:
 - Eleven cyclic paths in top 80% of all weights:
 $D = [0.23, 0.14, 0.077, \dots, 0.023, 0.014, 0.012]$
 - Thirteen non-cyclic paths & cyclic paths in bottom 20%
 of weights, combined weight = 0.30

High complexity case: score 6.44

Complexity from CN segments = $\log(84) = 4.43$

Complexity of non-ecDNA-like paths
 = $-0.30 \times \log(0.30) = 0.36$

Complexity of ecDNA-like paths
 = $-\sum D_i \times \log(D_i) = 1.65$

Total complexity score:
 = $4.43 + 0.36 + 1.65$
 = **6.44**

Additional points.

Referee 3, Point 6: "While this study is very comprehensive, the study should cite some of the previous research on double minutes in BE, especially where they are recognized for their clinical relevance (e.g., ERBB2, MYC, ZNF217)."

Response: Thank you for suggesting this. We have added the following sentence to the introduction: (line 80) “Prior reports have raised the possibility of ecDNA in Barrett’s-derived precancer samples suggesting a potential role for ecDNA malignant transformation to EAC.

The above sentence references Paulson et al., Nat. Commun. 2022 (PMID: 35484108), Ng et al., Commun. Biol. 2022 (PMID: 35396535), Stachler et al., biorXiv 2021, (<https://doi.org/10.1101/2021.03.26.437288>), and Nones et al., 2014 (PMID: 25351503) all of which identify ecDNA (sometimes termed “double-minutes”) in their analyses.

Minor comments:

Referee 3, Point 7: “While the importance of the ecDNA-2 associated genes (SOCS1, CIITA, RMI2) in P391 is very convincing, are these also highly expressed? Might the authors potentially have access to any histological for IHC staining of these genes to demonstrate the expression of these genes?”

Response: Regrettably, no RNA-seq data was generated for the FHCC samples and no tissue was available from the biopsies, which were collected throughout the 1990’s.

We have, however, previously shown that gene encoded on ecDNA, especially oncogenes, are associated with high-level mRNA expression (Wu et al., Nature 2019, PMID: 31748743) and even high levels of protein expression of genes encoded on ecDNAs (Lange et al., Nature Genetics, 2022). Future studies will be needed to examine the expression of immunomodulatory genes on ecDNAs.

However, to lend further confidence to the reviewer, we analyzed expression data from genes encoded on ecDNA in HGD in a UK sample, as follows:

The figure above shows in grey, the expression of genes in the HGD ecDNA region for the HGD sample (red) and for 6 patients having normal squamous esophagus (NE), gastric cardia (GC) and duodenum (DU), each. We detected elevated gene expression in the copy-number amplified regions, compared to normal samples, including for oncogenes such as *STAT3* and *ERBB2*.

Referee 3, Point 8: “The number of the Cambridge cohort seems to be wrong and incomplete in the method section. Should it be 51 early-stage T1, plus 88 late stage ECAs (line 302)?”

Response: We thank the reviewer for pointing out this error and have corrected the numbers in the revised MS lines 342-346.

Referee 3, Point 9: “The phrase ‘highly significant’ on line 129 should be changed or a p-value provided (e.g., 0.0003 for one-sided NCO vs CO).”

Response: We have changed the text as suggested in the revised MS, adding in the p-value. The manuscript now reads (line 143):

“Furthermore, in both timepoints together, ecDNA was found in samples from 1/40 NCO patients and in samples from 13/40 CO patients (Figure 2d), demonstrating a highly significant association between ecDNA in BE biopsies and progression to EAC (Fisher’s exact test, $p=3.3 \times 10^{-4}$, one-sided test).”

Referee 3, Point 10: “How concordant did the authors find the purity estimates of the pathologist and those determined by sequencing (line 326)?”

Response: Please see detailed answer to point 1, above. To summarize, the histopathology estimates are widely acknowledged to be an underestimate, which is why we choose a high stringency for the Cambridge cohort. For the FHCC cohort, unfortunately, pathology wasn’t available on the sequenced tissue, as described above.

Referee 3, Point 11: “What was the archived coverage, purity & ploidy distribution for WGS samples across the different disease stages and cohorts?”

Response: We have provided the table below, as Supplementary Table 4 of the revised MS, summarizing these data (summary shown below). Graphic summaries are also provided below for the reviewers’ convenience. Please also see detailed response to Referee 3, Point 1, above.

Feature	Source	Mean values
Sequencing coverage (FHCC)	Sequencing data	73.5 (NCO), 74.1 (CO)
Ploidy (FHCC)	Sequencing data	2.03 (NCO), 2.28 (CO)
Ploidy (Cambridge)	Sequencing data	2.22 (BE), 2.35 (LGD), 2.65 (HGD), 2.71 (Early EAC), 2.88 (Late EAC)
Sample purity (FHCC)	Sequencing data	0.88 (NCO), 0.83 (CO)
Aberrant cell fraction (Cambridge)	Sequencing data	0.54 (BE), 0.56 (LGD), 0.47 (HGD), 0.53 (Early EAC), 0.48 (Late EAC)
Cellularity percentage	Histology	67% (BE), 55% (LGD), 49% (HGD), 71% (Early EAC), 82% (Late EAC)

We thank the Referee 3 for these highly constructive and insightful comments. Addressing them has strengthened the manuscript.

Reviewer Reports on the First Revision:

Referees' comments:

Referee #1 (Remarks to the Author):

The authors have addressed not only my comments and questions adequately, but also those of the other two reviewers in my opinion. They have provided ample, and even at times significant additional, analysis and data to support their responses to the reviewers' comments and questions.

Referee #2 (Remarks to the Author):

In this revised manuscript, Luebeck et al have appropriately responded to reviewer critiques by changes to the text, generation of additional data, and detailed explanations.

Using multiple well-characterized cohorts, they have demonstrated that ecDNA is associated with high grade dysplasia and EAC but not NDBE and LGD. Further, higher stage EAC patients had a higher prevalence of ecDNA. Temporal findings in a single patient correlate progression from HGD to EAC with the acquisition of a second ecDNA. These findings suggest that ecDNA play a role in EAC pathogenesis.

These findings are original and significant.

There was appropriate use of statistics and presentation of error bars and indication of significance.

Overall, the manuscript was well written.

Referee #3 (Remarks to the Author):

The authors appropriately answered my comments, I am happy with the additional analysis and results.